# Bidirectional Decoding: Improving Action Chunking via Guided Test-Time Sampling

**Yuejiang Liu,**\* **Jubayer Ibn Hamid,**\* **Annie Xie, Yoonho Lee, Maximilian Du, Chelsea Finn**
Department of Computer Science, Stanford University

## Abstract

Predicting and executing a sequence of actions without intermediate replanning, known as action chunking, is increasingly used in robot learning from human demonstrations. Yet, its effects on the learned policy remain inconsistent: some studies find it crucial for achieving strong results, while others observe decreased performance. In this paper, we first dissect how action chunking impacts the divergence between a learner and a demonstrator. We find that action chunking allows the learner to better capture the temporal dependencies in demonstrations but at the cost of reduced reactivity to unexpected states. To address this tradeoff, we propose *Bidirectional Decoding* (BID), a test-time inference algorithm that bridges action chunking with closed-loop adaptation. At each timestep, BID samples multiple candidate predictions and searches for the optimal one based on two criteria: (i) backward coherence, which favors samples that align with previous decisions; (ii) forward contrast, which seeks samples of high likelihood for future plans. By coupling decisions within and across action chunks, BID promotes both long-term consistency and short-term reactivity. Experimental results show that our method boosts the performance of two state-of-the-art generative policies across seven simulation benchmarks and two real-world tasks. Videos and code are available at `https://bid-robot.github.io`.

## 1 Introduction

The increasing availability of human demonstrations has spurred renewed interest in behavioral cloning (Atkeson & Schaal, 1997; Argall et al., 2009). In particular, recent studies have highlighted the potential of learning from large-scale demonstrations to acquire a variety of complex skills (Zhao et al., 2023b; Chi et al., 2023; Fu et al., 2024b; Lee et al., 2024; Khazatsky et al., 2024). Yet, existing methods still struggle with two common properties of human demonstrations: (i) strong temporal dependencies across multiple steps, such as idle pauses (Chi et al., 2023) and latent strategies (Xie et al., 2021; Ma et al., 2024), (ii) large style variability across different demonstrations, such as differences in proficiency (Belkhale et al., 2024) and preference (Kuefler & Kochenderfer, 2017). Often, both properties are prevalent yet unlabeled in collected data, posing significant challenges to the traditional behavioral cloning that maps an input state to an action.

In response to these challenges, recent works have pursued a generative approach equipped with action chunking: (i) predicting a sequence of actions over multiple time steps and executing all or part of the sequence (Zhao et al., 2023b; Chi et al., 2023); (ii) modeling the distribution of action chunks and sampling from the learned model in an independent (Chi et al., 2023; Prasad et al., 2024) or weakly dependent (Janner et al., 2022; Zhao et al., 2023b) manner for sequential decisions. Some studies find this approach crucial for learning a performant policy in laboratory scenarios (Zhao et al., 2023b; Chi et al., 2023), while other recent work reports opposite outcomes under practical conditions (Lee et al., 2024). The reasons behind these conflicting observations remain unclear.

In this paper, we first dissect the influence of action chunking by examining the divergence between learned policies and human demonstrations. We find that, when a policy is built with limited context length – little or no history is used as input for robustness or efficiency (Mandlekar et al., 2020; Bharadhwaj et al., 2023b; Brohan et al., 2023a;b; Shi et al., 2023; Collaboration, 2023) – increasing the length of action chunks allows for implicit conditioning on more past actions, thereby improving its ability to capture the temporal dependencies inherent in demonstrations. However, this advantage

---

\*Equal contribution. Correspondence to yuejiang.liu@stanford.edu. Videos at https://bid-robot.github.io.

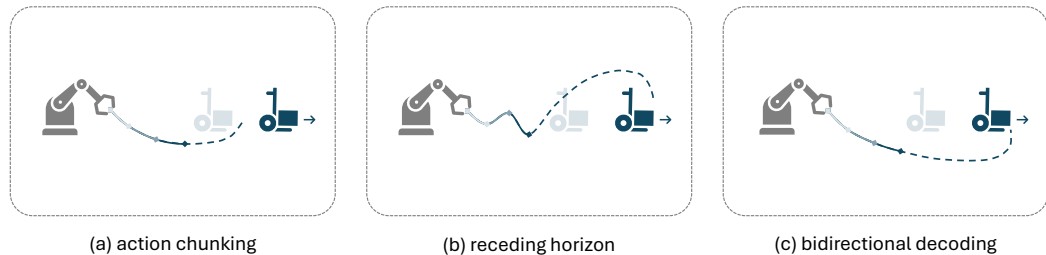

Figure 1: Illustration of different inference methods applied to a robot policy with action chunking. The robot is tasked with catching a moving trolley. (a) Vanilla action chunking (Zhao et al., 2023b) executes actions based on previous predictions, resulting in delayed reactions to object motions. (b) Receding horizon (Chi et al., 2023) enables faster reactions, but leads to a jittery trajectory in the presence of multimodal demonstrations (*e.g.*, both left- and right-handers). (c) Our Bidirectional Decoding explicitly searches for the optimal action from multiple predictions sampled at each time step, achieving both long-term consistency and short-term reactivity.

comes at the cost of reduced access to recent state observations, which can be crucial for reacting to unexpected dynamics arising from modeling errors or environmental stochasticity. This trade-off raises a crucial question: How can we preserve the strengths of action chunking in long-term consistency without suffering from its limitations in short-term reactivity?

To this end, we introduce *Bidirectional Decoding* (BID), an inference algorithm that bridges action chunking with closed-loop adaptation. Our main idea is to sample multiple predictions at each time step and search for the most desirable one. Specifically, BID operates on two decoding criteria: (i) backward coherence, which favors samples that are close to the sequence selected at the previous step; (ii) forward contrast, which favors samples that are close to the output of a stronger policy and distant from those of a weaker one. As illustrated in Fig. 1, BID updates the chunk of future actions based on the previous strategy, promoting temporal consistency over extended periods while remaining reactive to unexpected changes.

The main contributions of this paper are twofold: (i) a thorough analysis of action chunking (§3), and (ii) a decoding algorithm to improve it (§4). Empirically, we validate our theoretical analysis through a one-dimensional diagnostic simulation and evaluate our decoding method on two state-of-the-art generative policies across seven simulations and two real-world tasks (§5). Our experiment results show that the proposed BID boosts the performance of recent policies by more than 32% in relative performance. BID is model-agnostic, computationally efficient, and easy to implement, serving as a plug-and-play component to enhance generative behavior cloning at test time.

## 2 RELATED WORK

**Behavioral Cloning.** Learning from human demonstrations is becoming increasingly popular in robot learning due to recent advances in robotic teleoperation interfaces (Sivakumar et al., 2022; Zhao et al., 2023b; Wu et al., 2023; Chi et al., 2024). Generative Behavior cloning, which models the distribution of demonstrations, is particularly appealing due to its algorithmic simplicity and empirical efficacy (Jang et al., 2022; Florence et al., 2022; Brohan et al., 2022; Shafiullah et al., 2022; Zhao et al., 2023b; Chi et al., 2024; Brohan et al., 2023a). However, a significant limitation is compounding errors, where deviations from the training distribution accumulate over time (Ross et al., 2011; Ke et al., 2021). These errors can be mitigated by gathering expert correction data (Ross et al., 2011; Kelly et al., 2019; Menda et al., 2019; Hoque et al., 2021a;b) or injecting noise during data collection (Laskey et al., 2017; Brandfonbrener et al., 2023), but such strategies require additional time and effort from human operators. To address this, recent work proposes predicting a sequence of multiple actions into the future, known as action chunking, which reduces the effective control horizon (Lai et al., 2022; Zhao et al., 2023b; George & Farimani, 2023; Bharadhwaj et al., 2023a). By handling sequences of actions, action chunking is also better at handling temporal dependencies in the data, such as idle pauses (Swamy et al., 2022; Chi et al., 2023) or multiple styles (Li et al., 2017; Kuefler & Kochenderfer, 2017; Gandhi et al., 2023; Belkhale et al., 2024). However, independently drawn action sequence samples may not preserve the necessary temporal dependencies for smooth and consistent execution. Our work provides a thorough analysis of action chunking and proposes a decoding algorithm to improve it.

**Sequential Decoding.** Decoding algorithms have been studied in generative sequence modeling for decades, with renewed attention driven by recent advances in large language modeling (LLM). One prominent approach focuses on leveraging internal metrics, *e.g.*, likelihood scores, to improve the quality of generated sequences. Notable examples include beam search (Freitag & Al-Onaizan, 2017; Vijayakumar et al., 2018), truncated sampling (Fan et al., 2018; Hewitt et al., 2022), minimum Bayes risk decoding (Kumar & Byrne, 2004; Müller & Sennrich, 2021), and others (Welleck et al., 2019; Meister et al., 2023; Fu et al., 2024a). Another line of research explores the distinctions between multiple models to jointly optimize for the desired properties such as quality or efficiency (Li et al., 2023; Leviathan et al., 2023). More recently, several studies have highlighted the potential of guiding the decoding or sampling process through the use of an external model, such as a classifier (Dhariwal & Nichol, 2021) or reward model (Khanov et al., 2023). In the context of robot learning, recent works have explored guided decoding for long-horizon robotic planning (Huang et al., 2023) and manipulator geometry designs (Xu et al., 2024). Nevertheless, effective decoding strategies for low-level robotic actions remain lacking. Concurrent to our work, Nakamoto et al. (2024) propose to select action samples by querying a value function learned from reward-annotated demonstrations (Hansen-Estruch et al., 2023). Our work does not rely on a separate value function; instead, we propose a decoding strategy that addresses the consistency-reactivity tradeoff inherent in action chunking through sample comparison.

## 3 ANALYSIS: TRADEOFFS IN ACTION CHUNKING

### 3.1 PRELIMINARIES

Consider a dataset of demonstrations $\mathcal{D} = \{\tau_i\}_{i=1}^N$, where each demonstration $\tau_i$ consists of a sequence of state-action pairs $\tau_i = \{(s_1, a_1), (s_2, a_2), \cdots, (s_T, a_T)\}$ provided by a human expert. At each time step $t$, the demonstrated action $a_t$ is influenced not only by the observed state $s_t$, but also by latent variables $z_t$, such as planning strategies (*e.g.*, subgoals) and personal preferences (*e.g.*, handedness). These latent variables can persist across multiple time steps and vary significantly between different demonstrations. Fig. 2 illustrates the decision process of a human expert, highlighting the inherent temporal dependencies.

To model these temporal dependencies, some recent works (Zhao et al., 2023b; Chi et al., 2023; Lee et al., 2024) utilize action chunking, *i.e.*, modeling the joint distribution of future actions conditioned on past states $\pi(a_t, a_{t+1}, \cdots, a_{t+l}|s_{t-c}, \cdots, s_t)$, or in short $\pi(a_{t:t+l}|s_{t-c:t})$. Here, $c$ is the *context length*, *i.e.*, number of past steps for state inputs, and $l$ is the *prediction length*, *i.e.*, number of future steps for action outputs. Training such a policy typically involves minimizing the divergence of action distributions between the model $\pi$ and the expert $\pi^*$,

$$\pi = \arg\min_\pi \sum_{\tau \in \mathcal{D}} \sum_{\substack{s_{t-c:t} \\ a_{t:t+l}}} \mathcal{L}(\pi(a_{t:t+l}|s_{t-c:t}), \pi^*(a_{t:t+l}|s_{t-c:t})). \tag{1}$$

During deployment, the policy operates with a specific *action horizon* $h \in [1, l]$, *i.e.*, executing part or all of the predicted action sequence for $h$ time steps without re-planning. This approach essentially takes in $c$ states as context and executes $h$ actions, which we refer to as a $(c, h)$-policy.

The choices of context length $c$ and action horizon $h$ often play a crucial role in the effectiveness of the learned policy. Recent policies often use a short context length $c$, as extending the context can lead to performance degradation in the presence of limited training data (refer to Appendix A.2 for more details). Conversely, extending the action horizon $h$ has produced mixed results. Some studies report benefits in laboratory settings (Zhao et al., 2023b; Chi et al., 2023), while others find it detrimental in real-world scenarios (Lee et al., 2024). The reasons behind these conflicting outcomes are not well understood yet.

Notably, Zhao et al. (2023b) hypothesize that action chunking mitigates compounding errors, but it is unclear why this would hold when deviations within a chunk cannot be corrected before replanning. Alternatively, Chi et al. (2023) draws parallels to model predictive control (MPC) (Borrelli et al., 2017; Löfberg, 2012), emphasizing its role in improving consistency and planning. However, unlike MPC, which typically uses short action horizons (*e.g.*, $h = 1$), recent imitation learning methods employ much longer action horizons, such as a substantial portion of the predicted sequence (*e.g.*, $h = 8$ in Chi et al. (2023)) or even the entire prediction length (*e.g.*, $h \geq 50$ in Zhao et al. (2023b); Black et al. (2024)). The lack of a clear understanding of action chunking hinders its effective use

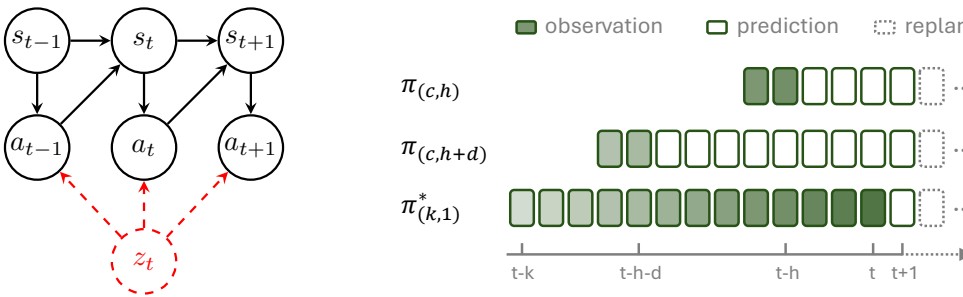

Figure 2: Illustration of the expert decision process, where a latent variable introduces temporal dependencies in actions.

Figure 3: Illustration of $(k, 1)$-expert, $(c, h)$-learner, and $(c, h + d)$-learner. Shaded regions represent observed history; darker indicate greater influence on current decision.

across tasks and environments. We next explicitly analyze the strengths and weaknesses of action chunking, with particular attention to the choice of action horizon $h$.

## 3.2 ANALYSIS

To understand the influence of action chunking, we focus on the last time step of the executed chunk, where the discrepancy between the expert policy and the learned policy is most pronounced. At this time step $t$, a $(k, 1)$-expert written as $\pi^* \coloneqq \pi^*(a_t | s_{t-k:t}, z_{t-k:t})$ predicts $a_t$ by conditioning on $k$ steps of the past states and the corresponding latent variables. In contrast, a $(c, h)$-learner written as $\pi_{(c,h)} \coloneqq \pi_{(c,h)}(a_t \mid s_{t-h-c:t-h}, a_{t-h:t-1})$ is constrained to observe $c$ steps of the past states and $h - 1$ steps of the past actions within the predicted chunk.

The divergence between a learned policy and an expert policy is attributed to two factors: (i) the importance of unobserved states in predicting the current action, and (ii) the difficulty of inferring unobserved states based on the available information.

To more clearly see the influence of action horizon on these factors, we next compare the performance of two policies that have the same context lengths but different action horizons, $\pi_h \coloneqq \pi_{(c,h)}(a_t | s_{t-h-c:t-h}, a_{t-h:t-1})$ and $\pi_{h+d} \coloneqq \pi_{(c,h+d)}(a_t | s_{t-h-d-c:t-h-d}, a_{t-h-d:t-1})$, where $d > 0$ is the extended action horizon. As illustrated in Fig. 3, each policy has access to unique information that is unavailable to the other. $\pi_h$ observes some recent states, where $\pi_{h+d}$ is only aware of the executed actions. On the other hand, $\pi_{h+d}$ has access to some earlier states and actions, which precede all information available to $\pi_h$. We characterize the *importance* of observations as follows (formal definitions in Appendix E.2):

**Definition (Expected Observation Advantage).** If a policy can observe a state $s_t$, we say that it has an observation advantage $\alpha_t$ over another policy that cannot observe it. More formally, this is the difference between the expected divergence accumulated by a policy $\pi(a_{t'} | s_{t'})$ that does not condition on the state $s_t$ and that by a policy $\pi(a_{t'} | s_{t'}, s_t)$ that conditions on $s_t$.

**Definition (Maximum Inference Disadvantage).** If a policy cannot condition its prediction on a state $s_t$, its maximum inference disadvantage $\epsilon_t$ is the largest possible divergence arising from inferring incorrectly. Here, the maximum is taken over all possible incorrect inferences of $s_t$.

Hence, we denote the observation advantage that $\pi_h$ gains from the observed recent states by $\alpha_f$ and the maximum inference disadvantage it incurs from the earlier unobserved states by $\epsilon_b$, whereas $\pi_{h+d}$ conversely gains $\alpha_b$ but incurs $\epsilon_f$.

Let $P(s_{t+1} \mid s_t, a_t)$ denote the true transition probabilities and $\hat{P}(s_{t+1} \mid s_t, a_t)$ the implicit dynamics model learned by the policy. The *difficulty* of inferring an unobserved state hinges on both the relevant observations and environmental stochasticity, which we quantify as follows.

**Definition (Forward Inference).** Let $P_f(t) \coloneqq P(S_t = g_t | S_{t-1} = g_{t-1}, A_{t-1} = a_{t-1})$, where $g_t$ and $g_{t-1}$ are the ground truth states at time $t$ and $t - 1$ respectively in a deterministic environment. In deterministic environments, $P_f(t) = 1$; whereas in stochastic settings, $P_f(t)$ is smaller. The prediction error is characterized by $\delta_f(t) = \hat{P}(S_t = g_t | S_{t-1} = g_{t-1}, a_{t-1}) - P(S_t = g_t | S_{t-1} = g_{t-1}, a_{t-1})$.

**Definition (Backward Inference).** Similarly, let $P_b(t) \coloneqq P(S_t = g_t | S_{t+1} = g_{t+1})$ where $g_t$ and $g_{t+1}$ are the ground truth states in the deterministic environment at time $t$ and $t + 1$, respectively.

Since $P_b(t)$ is not conditioned on any action, it has higher entropy in general. In stochastic environments, $P_b(t)$ is small. Furthermore, let $\delta_b(t) = \hat{P}(S_t = g_t | S_{t+1} = g_{t+1}) - P(S_t = g_t | S_{t+1} = g_{t+1})$.

We refer to the variance of the distribution $P(a_{t'-1}, \ldots, a_{t'-d} \mid s_{t'})$ under expert policy as the diversity of past strategies up to time $t'$. Given that the forward inference is generally easier than the backward inference, the performances of $\pi_h$ and $\pi_{h+d}$ differ as follows (proofs are deferred to Appendix E):

**Proposition 1** (Consistency-Reactivity Inequalities). *Let $\mathcal{L}$ be a non-linear and non-negative convex function measuring the prediction error with respect to demonstrations. Let $C := \{a_{t-h:t-1}\} \cup \mathcal{S}^+$ where $\mathcal{S}^+$ are the common states that both $\pi_h$ and $\pi_{h+d}$ observe. For the ease of notation, let $\tau_f = \{t - h - d + 1 : t - h\}$ and let $\tau_b = \{t - h - d - c : t - h - c - 1\}$. Then, the difference in the expected loss between the $(c, h + d)$-policy and the $(c, h)$-policy, $\Delta_d := \min_{\pi_{h+d}} \mathbb{E}[\mathcal{L}(\pi_{h+d}, \pi^*) | C] - \min_{\pi_h} \mathbb{E}[\mathcal{L}(\pi_h, \pi^*) | C]$, is bounded as:*

$$\alpha_f - \epsilon_b \left(1 - \prod_{\tau \in \tau_b} P_b(\tau)(P_b(\tau) + \delta_b(\tau))\right) \leq \Delta_d \leq \epsilon_f \left(1 - \prod_{\tau \in \tau_f} P_f(\tau)(P_f(\tau) + \delta_f(\tau))\right) - \alpha_b. \tag{2}$$

*Remark 1.* Eq. (2) provides a general comparison of the performance of the two policies. Intuitively, the advantage of each policy stems from the additional information it has access to (*i.e.* $\alpha_f$ for $\pi_h$ and $\alpha_b$ for $\pi_{h+d}$) while the disadvantage is bounded by the maximum divergence arising from inferring missing information incorrectly (*i.e.* $\epsilon_b$ and $\epsilon_f$ scaled by the maximum probability of incorrect inference).

We next examine the implications of Proposition 1.

**Corollary 2** (Consistency). *Suppose the train and test environments are identical and deterministic. Suppose $a_t$ is influenced by at least one state at time steps $\tau_b$ and let $\delta_f(t') \approx 0$ for all $t' \in \tau_f$. If the diversity in past strategies up to time $t - h - c$ is not 0, and $\epsilon_f$ is finite, then*

$$\min_{\pi_{h+d}} \mathbb{E}[\mathcal{L}(\pi_{h+d}, \pi^*) | C] < \min_{\pi_h} \mathbb{E}[\mathcal{L}(\pi_h, \pi^*) | C]. \tag{3}$$

*Remark 2.* In deterministic environments, while both policies need to infer the same number of unobserved states, $\pi_{h+d}$ benefits from conditioning on additional actions, which may significantly aid in inferring the corresponding states through its action chunk. However, this is only true if the implicit dynamics model learned by the policy is approximately accurate and the maximum errors $\epsilon_f$ arising from inferring these states are bounded. Note that $\pi_{h+d}$ performs better as we increase the diversity of strategies present in the dataset *i.e.*, as the distribution of actions taken by the expert prior to the context of the short horizon policy becomes higher variance.

**Corollary 3** (Reactivity). *Suppose $P_f(t')$ is small or both $P_f(t')$ and $|\delta_f(t')|$ are large for all $t' \in \tau_f$. If temporal dependency decreases over time such that $\epsilon_b$ is small and $a_t$ is influenced by at least one state at time steps in $\tau_f$, then*

$$\min_{\pi_{h+d}} \mathbb{E}[\mathcal{L}(\pi_{h+d}, \pi^*) | C] > \min_{\pi_h} \mathbb{E}[\mathcal{L}(\pi_h, \pi^*) | C]. \tag{4}$$

*Remark 3.* Recent states are often more important to decision making and, therefore, the disadvantages of action chunking become pronounced when inferring these recent states becomes difficult. This can happen if the test environment is stochastic. This can also happen if the test environment is deterministic as long as the implicit dynamics model is inaccurate - either due to a distribution shift or because of learning difficulty.

In summary, depending on the experimental conditions, action chunking may have varying effects on the learned policy. On the one hand, it benefits the modeling of temporal dependencies in the demonstrations, due to extended access to actions executed at past time steps. On the other hand, it hinders reactions to unexpected dynamics, due to reduced access to state observations at recent time steps. As a result, there is no universally optimal choice of action horizon across all conditions. When both temporal dependencies and prediction errors are significant, tuning the action horizon entails an inherent trade-off between the two opposing factors.

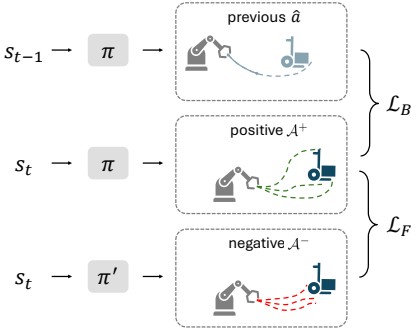

$s_{t-1} \rightarrow \pi \rightarrow$ previous $\hat{a}$

$s_t \rightarrow \pi \rightarrow$ positive $\mathcal{A}^+$

$s_t \rightarrow \pi' \rightarrow$ negative $\mathcal{A}^-$

$\mathcal{L}_B$

$\mathcal{L}_F$

Figure 4: Illustration of bidirectional decoding.

---

**Algorithm 1** Bidirectional Decoding

**Require:** current state $s$, batch size $N$, mode size $K$, previous decision $\hat{a}$, strong policy $\pi$, weak policy $\pi'$
1: Generate $N$ samples from each policy $a \sim \pi(s)$, $a' \sim \pi'(s)$ to construct the initial sets $\mathcal{A}$ and $\mathcal{A}'$
2: Compute the backward loss $\mathcal{L}_B$ for each sample
3: Select K samples with minimal $\mathcal{L}_B$ from $\mathcal{A}$ and $\mathcal{A}'$ to construct $\mathcal{A}^+$ and $\mathcal{A}^-$, respectively
4: Compute the forward loss $\mathcal{L}_F$ for each sample
5: Select $a^* \in \mathcal{A}$ that minimizes the total loss
6: Update decision memory $\hat{a} \leftarrow a^*$

---

## 4 METHOD: BIDIRECTIONAL DECODING

As analyzed above, action chunking improves long-term consistency but sacrifices short-term reactivity. In this section, we propose to address this tradeoff by bridging action chunking with closed-loop adaptation. We will first outline the general framework in §4.1 and then describe two specific criteria in §4.2.

### 4.1 TEST-TIME SEARCH

Recall that for a policy with prediction length $l$, an action chunk $a := \{a_t^{(t)}, a_{t+1}^{(t)}, \cdots, a_{t+l}^{(t)}\}$ sampled at time $t$ is expected to follow a consistent strategy over the subsequent $l$ time steps. However, executing the action chunk in an open-loop manner leaves the policy vulnerable to unexpected state changes. Alternatively, one can maximize reactivity by resampling an action chunk and executing only the first immediate action at every time step. Yet, this simple closed-loop approach destroys the consistency preserved within each chunk, potentially leading to oscillations between different strategies. How can we combine the benefits of both approaches?

Our key idea is to bridge action chunking and closed-loop resampling by making use of additional computes at test time. In particular, we seek to restore temporal consistency in closed-loop operations by scaling up the number of candidate samples. Intuitively, while the probability of any single pair of samples sharing the same latent strategy is low, the likelihood of finding a consistent pair increases with the number of samples. We thus frame the problem of closed-loop action chunking as searching for the optimal action among a batch of samples drawn at each time step,

$$a^* = \arg\min_{a \in \mathcal{A}} \mathcal{L}_B(a) + \mathcal{L}_F(a), \tag{5}$$

where $\mathcal{A}$ is the set of sampled action chunks, $\mathcal{L}_B$ and $\mathcal{L}_F$ are two criteria approximating the optimality with respect to the backward decision and forward plan, which we will describe next.

### 4.2 BIDIRECTIONAL CRITERIA

**Backward coherence.** To preserve temporal dependencies in closed-loop operations, a sequence of actions should (i) commit to a consistent latent strategy over time, and (ii) react smoothly to unexpected changes. These desired properties motivate us to keep the action chunk selected at the previous time $\hat{a} := \{a_{t-1}^{(t-1)}, \cdots, a_{t+l-1}^{(t-1)}\}$ as a prior, and minimize the weighted Euclidean distance between the new action chunk and the prior across $l - 1$ overlapping steps:

$$\mathcal{L}_B = \sum_{\tau=0}^{l-1} \rho^\tau \left\| a_{t+\tau}^{(t)} - a_{t+\tau}^{(t-1)} \right\|_2. \tag{6}$$

Here, $\rho$ is a decay hyperparameter to account for growing uncertainty over time. By default, we use the L2 norm as the distance metric between predicted actions, while other metrics, such as L1 or cosine distance, can also be effective (see Appendix A.6). This backward objective encourages similar latent strategies between consecutive steps while allowing for gradual adaptation to unforeseen transition dynamics.

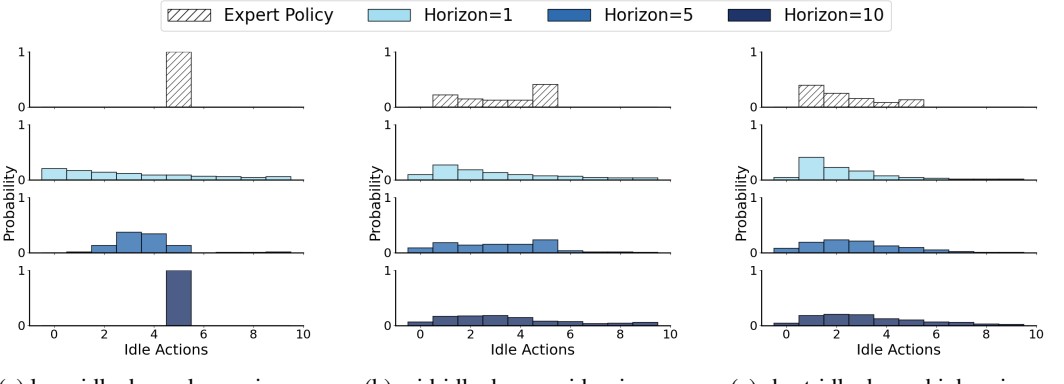

Figure 5: Effect of action horizon $h$ on idle actions in 1-dimensional simulations. All policies share the same prediction length $l$. Long action horizons lead to idle distributions closer to the long-idle expert in low-noise environments, whereas shorter action horizons align more closely with the short-idle expert in high-noise environments. When both idling and noise are non-negligible, a moderate action horizon performs the best.

Nevertheless, the backward criterion alone presents a potential caveat: the prior chunk could be suboptimal due to the lack of information at the previous time step (*e.g.*, unexpected object motions). In such cases, selecting the next action chunk based solely on the prior may perpetuate suboptimality. Ideally, the sequential decision-making process should effectively correct suboptimal plans based on the latest observations. We next address this need through another forward criterion.

**Forward contrast.** Our design of the forward criterion is motivated by the need to identify the most optimal plan from a set of candidates. Within the same latent strategy, suboptimal samples may arise from (i) low likelihood under the learned model and (ii) divergence between the learned model and expert policy. To address this, we draw inspirations from LLM decoding techniques (Wang et al., 2022; Li et al., 2023) and introduce a forward contrast criterion. Specifically, we compare each candidate sample with two sets of reference samples: one positive set from a stronger policy and a negative set from a weaker one. The stronger policy is obtained from a well-trained checkpoint, whereas the weaker policy is taken from an early underfitting checkpoint and is expected to be further from the expert policy. Our forward objective is thus framed as minimizing the average distance between a candidate plan and the positive samples while maximizing its average distance from the negative ones,

$$\mathcal{L}_F = \frac{1}{N} \left( \sum_{a^+ \in \mathcal{A}^+} \sum_{\tau=0}^{l} \left\| a_{t+\tau}^{(t)} - a_{t+\tau}^+ \right\|_2 - \sum_{a^- \in \mathcal{A}^-} \sum_{\tau=0}^{l} \left\| a_{t+\tau}^{(t)} - a_{t+\tau}^- \right\|_2 \right), \qquad (7)$$

where $\mathcal{A}^+ = \mathcal{A} \setminus \{a\}$ is the positive set predicted by the strong policy $\pi$, $\mathcal{A}^-$ is the negative set predicted by the weaker one $\pi'$, and $N$ is the sample size.

Fig. 4 illustrates the combined effects of the backward coherence and forward contrast criteria on sample selection. Since samples in $\mathcal{A}^+$ and $\mathcal{A}^-$ are not all subject to the same strategy, we trim each set by removing samples that deviate significantly from the previous decision. This is achieved by summing over the $K$ smallest distance values for in the positive and negative sets in Eq. (7). The full process of our decoding method is outlined in Algorithm 1. Since all steps in BID can be computed in parallel, the overall computational overhead remains modest on modern GPU devices. Please refer to Appendix B for additional discussions about our decoding method.

## 5 EXPERIMENTS

In this section, we present a series of experiments to answer the following questions:

1. How does our theoretical analysis on action chunking manifest under different conditions?
2. Can our decoding method improve closed-loop operations of policies built with action chunking?
3. Does our decoding method perform well across different policies, tasks, and environments?
4. Is our decoding method scalable to larger sample sizes and compatible with existing methods?

To this end, we will first validate our theoretical analysis through one-dimensional diagnostic simulations. We will then evaluate BID on seven tasks across three simulation benchmarks, including

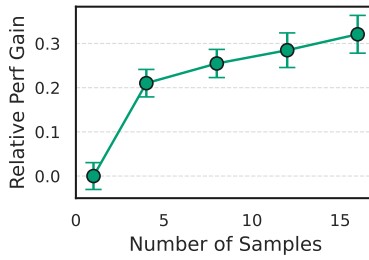

Figure 6: Comparison of different inference methods for closed-loop operations of diffusion policies. Each method is evaluated for 100 episodes on seven manipulation tasks in simulation benchmarks. Results are averaged across three seeds. BID significantly outperforms existing inference methods in most tasks.

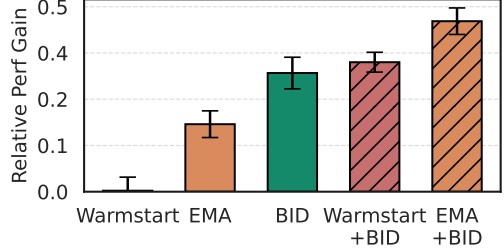

Figure 7: BID benefits from large sample sizes (left) and complements existing inference methods (right). Each method is evaluated on seven simulation tasks across three seeds. Relative performance gain is measured with respect to the vanilla baseline. When combined with EMA, BID results in 46% relative improvements.

Push-T (Chi et al., 2023), RoboMimic (Mandlekar et al., 2022), and Franka Kitchen (Gupta et al., 2020). We will subsequently examine the generality and scalability of our method under various base policies, sample sizes, and stochastic noise. We will finally assess the effectiveness of BID in two challenging real-world tasks that require interactions with dynamic objects.

## 5.1 ONE-DIMENSIONAL DIAGNOSTIC EXPERIMENTS

**Setup.** We start with a diagnostic experiment in a one-dimensional state space $\{s_0, s_1, \cdots, s_{10}\}$, where $s_0$ is the starting state and $s_{10}$ is the goal state. The demonstrator plans to move forward by one step in each state, except in $s_5$ where it pauses unless the last five states visited were $s_5$. Each forward move has a probability of $1 - \delta$, where $\delta$ denotes the level of stochastic noise in the environment (as described in §3.2). Given these demonstrations, we train a collection of policies with different action horizons $h \in \{1, 2, 3, 5, 7, 10\}$. We investigate under what action horizon our learner can better imitate the distribution of idle actions taken by the expert over multiple rollouts.

**Result.** As shown in Fig. 5, when the environment is deterministic ($\delta = 0.0$), larger action horizons capture the expert distribution better, consistent with Corollary 2. With an action horizon of 10, the learner achieves zero total variation distance with the expert distribution. Conversely, when the environment is highly stochastic ($\delta = 0.4$), an action horizon of 1 outperforms all other learners, corroborating with Corollary 3. With moderate noise ($\delta = 0.2$), there is no monotonic pattern and the optimal policy turns out to be the one with action horizon 5. Refer to Appendix A.1 for more detailed results. This controlled experiment validates the tradeoff identified in Proposition 1, which we will further demonstrate in robotic manipulation subjected to stochastic noise in §5.2.3

## 5.2 SIMULATION EXPERIMENTS WITH STOCHASTIC NOISE

Next, we evaluate our decoding algorithm on seven simulation tasks of robot manipulation. We will first compare BID with existing inference methods in closed-loop operations. We will then assess the effectiveness of our method under different conditions, including policy classes, sample sizes, and levels of stochastic noise.

### 5.2.1 COMPARISON WITH EXISTING INFERENCE METHODS

**Setup.** In each manipulation task, we use Diffusion Policy (Chi et al., 2023) trained on human demonstrations as the base policy. We evaluate BID with a batch size of $N = 16$ and a mode size of $K = 3$. We consider three existing inference methods as baselines:

- *Vanilla (Chi et al., 2023)*: Execute the first action of a sampled chunk in a closed-loop manner.

| Stochastic Noise | 0.0 | 1.0 | 1.5 |
|---|---|---|---|
| Vanilla Open-Loop | 64.0 | 26.9 | 13.0 |
| BID Open-Loop | **66.1** | 31.4 | 16.0 |
| Vanilla Closed-Loop | 48.9 | 38.3 | 29.5 |
| BID Closed-Loop | 54.4 | **45.3** | **31.7** |

| Sample Size | | Success (%) | Time (ms) |
|---|---|---|---|
| 1 | (vanilla) | 49.1 | 13.2 |
| 8 | (ours) | 52.9 | 25.2 |
| 16 | (ours) | 54.2 | 25.9 |
| 32 | (ours) | 54.4 | 26.8 |

Table 1: Success rates of VQ-BeT on the Push-T task under various noise conditions. Closed-loop BID substantially outperforms other methods in stochastic environments. See Table 4 for detailed ablations.

Table 2: Success rates and inference times of VQ-BeT across varying sample sizes. BID benefits from a larger sample size at the cost of a doubled computational overhead, measured on an A5000 GPU.

- *Warmstart (Janner et al., 2022)*: Similar to *Vanilla*, but warm-start the initial noise for the diffusion process from the previous decision.
- *Exponential Moving Average (EMA) (Zhao et al., 2023b)*: Smooth action chunking by averaging a new prediction $a$ with the previous one $\hat{a}$ for each overlapping step $a_t = \lambda a_t + (1 - \lambda)\hat{a}_t$. This method is also known as temporal ensembling. By default, we set $\lambda = 0.5$.

We evaluate each method for 100 episodes and average the results across three random seeds. Please refer to Appendix C for implementation details.

**Result.** Our main observation is that while existing inference methods offer some benefits for closed-loop operations, they lack robustness. As shown in Fig. 6, *Warmstart* yields mild performance gains on average, but degrades performance on 3 out of 7 tasks. Similarly, EMA leads to competitive results on several tasks, yet exhibits performance drops in 2 tasks. We conjecture that this robustness issue stems from independent sampling across chunks; when successive chunks follow distinct latent strategies, averaging them may not yield a plausible strategy, as further discussed in Appendix A.4. In comparison, BID offers substantial gains across all tasks. Notably, BID provides 32% relative improvements over the vanilla baseline, significantly outperforming EMA on Pust-T, Lift, Square, and Tool Hang, while achieving competitive performance on the other tasks.

### 5.2.2 SCALABILITY AND COMPATIBILITY OF BID

**Setup.** We further assess two key properties of BID: scalability with growing batch sizes and compatibility with existing inference methods. For scalability, we experiment with batch sizes of $\{1, 4, 8, 12, 16\}$. For compatibility, we apply BID on top of two other baselines, Warmstart and EMA. The results are averaged across seven simulation tasks and three random seeds.

**Result.** As shown in Fig. 7, our method benefits from the large batch size, with performance gains not yet saturated at the default batch size used in §5.2.1. Moreover, the strength from BID is complementary to that of existing inference methods. Notably, combining BID with EMA further boosts the relative performance gain from 32% to 46%. These two properties highlight the potential of our method in practice.

### 5.2.3 GENERALITY AND EFFICIENCY OF BID

**Setup.** We next extend our experiment to VQ-BET (Lee et al., 2024), another state-of-the-art robot policy built with autoregressive transformers. We use the public checkpoint on the Push-T task provided by LeRobot (Cadene et al., 2024) as the base policy. We use a checkpoint terminated at 100 epochs as the weak policy in forward contrast. To simulate stochastic conditions, we add temporally correlated Gaussian noise to the executed action at each step, scaled by the action magnitude. We measure the computational time on a desktop equipped with an NVIDIA A5000 GPU.

**Result.** Table 1 summarizes the results of the baseline and our method. The vanilla random sampling performs significantly worse than BID in both closed and open-loop operations. Notably, the vanilla open-loop approach exhibits a rapid performance decline as the environment becomes increasingly stochastic. Even in closed-loop operations, the vanilla baseline still experiences a significant performance drop. In comparison, the closed-loop BID demonstrates much higher robustness to stochastic noise. Table 2 details the computational overhead associated with BID at varying batch sizes. The result shows that the performance gains of our method come with a doubled computational overhead. We expect that this overhead will be less of a constraint with higher-end GPUs.

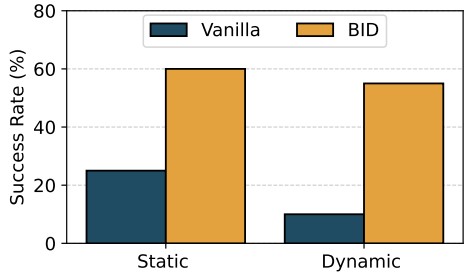 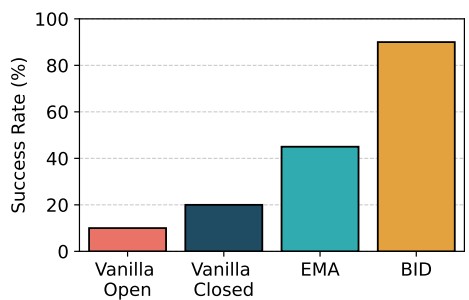

Figure 8: Success rate of object delivery. Each method-setting is evaluated across 20 episodes. BID achieves much higher success rate than the vanilla baseline, effectively handling the diverse demonstrations and dynamic target.

Figure 9: Success rate of cup replacement in the dynamic setting. Each method is evaluated across 20 episodes. Existing methods degrade substantially under slow cup movements, whereas BID retains a strong performance.

## 5.3 REAL-WORLD EXPERIMENTS WITH DYNAMIC OBJECTS

We finally evaluate BID through two real-world experiments that require rapid reactions to dynamically moving objects.

**Setup.** We consider two pick-and-place tasks, where the target object undergoes unexpected movement during evaluation. In the dynamic placing task, a Franka Panda robot is required to deliver an object into a moving cup held by a human subject. In the dynamic picking task, a UR5 robot is required to grasp a moving cup pulled by a string and place it onto a nearby static saucer. In both tasks, we evaluate the performance of BID applied to pre-trained diffusion policies. For further experimental details, please refer to Appendix C.2.

**Result.** Figs. 8 and 9 compares the results of different inference methods on the two real-world tasks. Vanilla random sampling struggles to handle the diverse demonstrations and dynamic movements, resulting in significantly lower success rates. In contrast, BID achieves high success rates in both static and dynamic conditions. Notably, in the dynamic picking task, BID achieves a 2x higher success rate than all other baselines, highlighting its potential for dynamic object interactions.

**Other experiments.**   Please refer to Appendix A for additional analyses and ablations.

## 6 CONCLUSION

**Summary.** We have analyzed the strengths and limitations of action chunking for robot learning from human demonstrations. Based on our analysis, we proposed Bidirectional Decoding (BID), an inference algorithm that takes into account both past decisions and future plans for sample selection. Our experimental results show that BID can consistently improve closed-loop operations, scale well with computational resources, and complement existing methods. We hope these findings provide a new perspective on addressing the challenges of generative behavioral cloning at test time.

**Limitations.** One major limitation of BID lies in its computational complexity. While the decoding computation can be fully parallelized on modern GPUs, it may remain expensive for high-frequency operations on low-cost robots. Designing algorithms that can generate quality yet diverse action chunks under batch size constraints can be an interesting avenue for future research. Additionally, our analysis and method have been limited to policies with short context lengths, driven by their empirical effectiveness with limited human demonstrations. Developing techniques capable of learning robust long-context policies can be another compelling direction for future research.

## REPRODUCIBILITY STATEMENT

Our code for the experiments with Diffusion Policy is available at `https://github.com/YuejiangLIU/bid_diffusion` and our code for the experiments with VQ-BET is available at `https://github.com/Jubayer-Hamid/bid_lerobot`. Additionally, detailed descriptions of our experimental setups are available in Appendix C, and complete proofs of our theoretical claims can be found in Appendix E.

ACKNOWLEDGMENTS

We would like to thank Anikait Singh, Eric Mitchell, Kyle Hsu, Moo Jin Kim, Annie Chen, Sasha Khazatsky, Rhea Malhotra and other members of the IRIS lab for valuable comments on this project. We would like to thank Yifan Hou, Huy Ha, Chuer Pan, Austin Patel, and the REAL lab for helpful support in setting up real-world experiments. We would like to thank Asher Spector and Danny Tse for their insightful feedback on the theoretical analysis. This work is supported by the AI Institute, ONR grant N00014-21-1-2685, an NSF CAREER award, and an SNSF Postdoc fellowship.

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

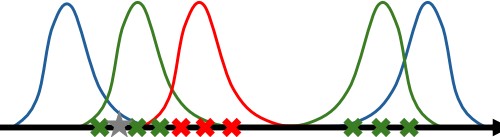

Figure 10: Distributional interpretation of BID. The backward criterion (Equation 6) favors samples close to the past decision; the forward criterion (Equation 7) promotes samples with a high likelihood under the target distribution.

| Noise Level | Action Horizon | | | | |
| --- | --- | --- | --- | --- | --- |
| | 1 | 3 | 5 | 7 | 10 |
| 0.0 | 4.03 | 2.07 | 1.54 | 1.06 | **0.00** |
| 0.2 | 1.43 | 0.94 | **0.39** | 0.71 | 1.32 |
| 0.4 | **0.36** | 0.42 | 0.59 | 0.835 | 1.11 |

Table 3: Total variation distance between the action distributions of each model and the expert in environments with varying noise levels. Lower values indicate better performance.

## A    ADDITIONAL EXPERIMENTS

### A.1    ONE-DIMENSIONAL SIMULATIONS

In addition to Fig. 5, we summarize the total variation distance between each learned policy and the demonstration in the one-dimensional simulation. Our results indicate that a shorter action horizon is more effective in noisier environments, whereas a longer action horizon yields better performance in static environments.

### A.2    ACTION HORIZON VS. CONTEXT LENGTH

**Setup.** Our work builds on the premise that the action horizon is longer than the context length, as commonly designed for recent policies. While BID mitigates the inherent limitations of this design choice through test-time decoding, an important question remains: could extending the context length yield stronger policies? To understand this, we trained diffusion policies with varying combinations of action horizons and context lengths on the Push-T task. Specifically, we use a short context length ($c = 2$) and a short action horizon ($h = 2$) as our baseline, set the prediction length equal to the action horizon $l = h$, and incrementally increase these parameters to larger values $6, 10, 14$ to assess their impact.

**Result.** Fig. 11 compares the performance of the policy learned with different $\Delta h = h - c$. As expected, the policy with both a short action horizon and a short context length does not perform well, due to its limited capability to model long-range temporal dependencies. Interestingly, extending the context length initially boosts performance ($\Delta h = -4$), but this trend reverses as the context length becomes too long ($\Delta h \leq -8$), likely due to overfitting to an increased number of spurious features. In contrast, expanding the action horizon results in more robust performance improvements, validating its pivotal role in imitation learning from human demonstrations.

### A.3    ABLATION STUDY OF FORWARD CONTRAST

**Setup.** To understand the effect of forward contrast (Equation 7), we evaluate the full version of our method against three reduced variants in open-loop operations: *Vanilla* (without forward contrast), *Positive* (without negative samples), and *Negative* (without positive samples). Similar to §5.2.2, our ablation study is conducted in seven simulation tasks, each with three random seeds.

**Result.** Fig. 12 summarizes the result of this ablation study. Notably, both positive and negative samples are essential for effective sample selection, and omitting either leads to significant performance declines. Without negative samples, our decoding method reduces to an approximate maximum a posteriori estimation, which can result in suboptimal decisions due to modeling errors. Conversely, without positive samples, the sampling process may be biased towards rare instances.

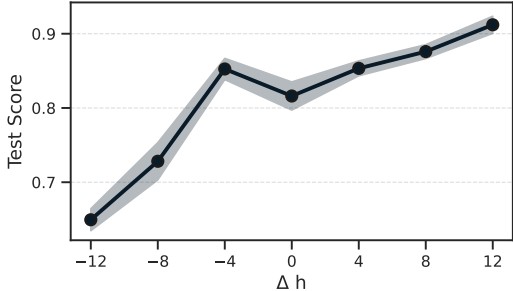

Figure 11: Effect of action horizon ($h$) and context length ($c$) on diffusion policies in the Push-T task. The baseline is set at $h = 2$ and $c = 2$, with $\Delta h = h - c = 0$. Extending the action horizon ($h > 2$) consistently improves performance, whereas extending the context length ($c > 2$) can cause substantial performance declines. Each model is trained for $5k$ epochs. Results are averaged over the last five checkpoints.

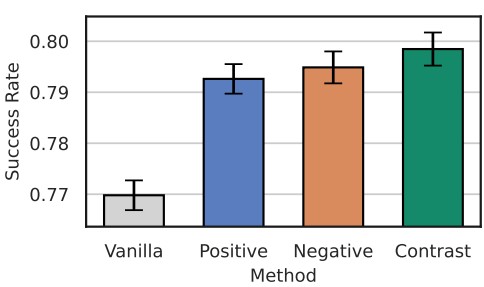

Figure 12: Effect of positive and negative samples on forward contrast. Performance of ablated variants of forward contrast is evaluated across seven simulation tasks. The absence of either positive or negative samples prevents achieving the full performance gains observed with the contrast objective.

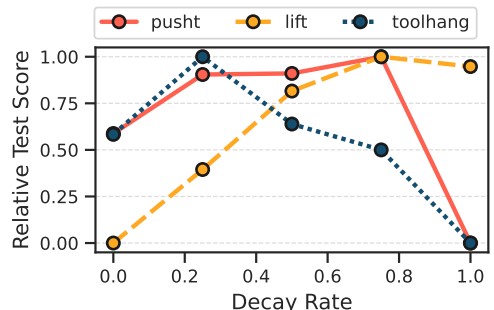

Figure 13: Effect of the decay rate for the exponential moving average. In each task, we measure the relative performance among different decay rates. The optimal decay rate varies by task, leading to a practical challenge of identifying a universal temporal ensembling strategy (Zhao et al., 2023b).

This result highlights the importance of both components and suggests the potential for extending this paradigm in future work.

## A.4 CHALLENGES FOR TEMPORAL ENSEBMLING

EMA exhibits competitive performance in Fig. 6. However, tuning its decay rate can be difficult in practice. Fig. 13 shows the sensitivity of EMA to the decay rate across three different tasks, where the optimal choices differ significantly. We conjecture that this high sensitivity stems from the variability in the latent strategies between consecutive predictions. When consecutive predictions follow similar strategies, a lower decay rate (*i.e.*, stronger moving average) can enhance smoothness and improve performance. Conversely, when consecutive predictions diverge in their underlying strategies, averaging them can introduce adverse effects. Our method promotes coherence in latent strategies and thus effectively complements temporal ensembling, as evidenced in Fig. 7.

## A.5 ABLATION STUDIES ON VQ-BET

Table 4 presents the performance of various sampling methods with VQ-BeT (Lee et al., 2024) on the Push-T task under different levels of environment stochasticity. We compare BID against vanilla sampling and EMA, for both open-loop and closed-loop executions. BID consistently outperforms the baselines, particularly in highly stochastic settings, with close-loop BID attaining the best performance on average. Moreover, the performance of EMA degrades significantly as the environment gets more noisy, likely due to the divergence in strategies between consecutive action chunks. Additionally, we conduct an ablation study on the forward contrast objective, isolating the roles of positive samples (from a strong model) and negative samples (from a weaker model). We observe

| Stochastic Noise | 0.0 | 1.0 | 1.5 | Average |
|---|---|---|---|---|
| Vanilla | $64.0 \pm 4.2$ | $26.9 \pm 2.8$ | $13.0 \pm 0.4$ | $34.6 \pm 1.7$ |
| EMA | $64.1 \pm 1.7$ | $27.6 \pm 3.3$ | $12.9 \pm 1.1$ | $34.9 \pm 1.3$ |
| Backward (ours) | $64.0 \pm 1.3$ | $27.6 \pm 1.0$ | $13.4 \pm 2.4$ | $35.1 \pm 1.0$ |
| + Positive (ours) | $65.6 \pm 1.9$ | $29.7 \pm 0.4$ | $15.7 \pm 1.7$ | $37.0 \pm 0.9$ |
| + Negative (ours) | $65.1 \pm 2.7$ | $26.6 \pm 0.8$ | $14.8 \pm 2.4$ | $35.5 \pm 1.2$ |
| BID (ours) | $\mathbf{66.1 \pm 3.5}$ | $\mathbf{31.4 \pm 3.0}$ | $\mathbf{16.0 \pm 1.2}$ | $\mathbf{37.8 \pm 1.6}$ |

(a) Open-Loop Operation

| Stochastic Noise | 0.0 | 1.0 | 1.5 | Average |
|---|---|---|---|---|
| Vanilla | $48.9 \pm 2.7$ | $38.3 \pm 3.4$ | $29.5 \pm 0.9$ | $38.9 \pm 1.5$ |
| EMA | $52.6 \pm 2.9$ | $35.7 \pm 2.2$ | $18.4 \pm 2.3$ | $35.6 \pm 1.4$ |
| Backward (ours) | $52.7 \pm 1.3$ | $42.0 \pm 3.2$ | $29.4 \pm 0.5$ | $41.4 \pm 1.2$ |
| + Positive (ours) | $53.8 \pm 3.3$ | $44.1 \pm 2.5$ | $30.4 \pm 0.8$ | $42.8 \pm 1.4$ |
| + Negative (ours) | $53.0 \pm 2.1$ | $44.3 \pm 2.2$ | $30.8 \pm 0.4$ | $42.7 \pm 1.0$ |
| BID (ours) | $\mathbf{54.4 \pm 1.8}$ | $\mathbf{45.3 \pm 3.8}$ | $\mathbf{31.7 \pm 0.3}$ | $\mathbf{43.8 \pm 1.4}$ |

(b) Closed-Loop Operation

Table 4: Success rates of VQ-BeT on the Push-T task under various conditions and sampling methods. The left table shows open-loop success rates, while the right table shows closed-loop success rates. BID consistently outperforms vanilla counterparts in both settings.

| | Vanilla | L2 (Ours) | L1 (Ours) | Cosine (Ours) |
|---|---|---|---|---|
| Square | $0.68 \pm 0.06$ | $\mathbf{0.76 \pm 0.03}$ | $\mathbf{0.76 \pm 0.04}$ | $0.73 \pm 0.04$ |
| Lift | $0.12 \pm 0.02$ | $0.58 \pm 0.06$ | $\mathbf{0.68 \pm 0.05}$ | $\mathbf{0.70 \pm 0.02}$ |
| Kitchen | $0.22 \pm 0.04$ | $0.64 \pm 0.05$ | $\mathbf{0.70 \pm 0.04}$ | $0.61 \pm 0.03$ |

Table 5: Effect of distance metric. We evaluate BID with different distance metrics on three robot manipulation tasks. Regardless of the chosen metric, BID substantially outperforms the vanilla baseline. Notably, the L1 distance yields an even stronger performance than the default L2 distance used in our other experiments.

that both types of samples contribute to the strong performance of BID across varying levels of environment stochasticity.

### A.6 Choice of Distance Metrics

As described in §4.2, our method uses the L2 distance as the default distance measure for quantifying similarity between action chunks. Yet, L2 distance is just one of several viable options. To assess the sensitivity of our approach to the choice of distance metric, we extend the experiments in §5.2.1 to further evaluate BID with alternative choices, including L1 and cosine distance. Results from three robot manipulation tasks, summarized in Table 5, demonstrate that BID achieves substantial performance gains regardless of the metric used. In particular, BID with L1 distance surpasses the default L2 distance in performance, highlighting the untapped potential of our method.

## B Additional Discussions

**Interpretation of our method.** Our method makes no changes to the learned policy; instead, it intervenes in the model distribution through sample selection. As illustrated in Fig. 10, randomly sampled sequences may be misaligned with both the previous decisions and the target demonstrations. Given a set of candidates, the backward step first identifies the behavioral mode from the past decision stored in memory; the forward step then removes the samples with low likelihood under the target distribution using prior knowledge of positive and negative samples. By comparing samples across time steps and model checkpoints, our method bridges the gap between the proposal and target distributions at test time.

**Relation to recent methods.** Our method builds upon the receding horizon (Chi et al., 2023) and temporal ensembling (Zhao et al., 2023b) used in previous works, but with crucial distinctions. Receding horizon seeks a compromise between long-term consistency and short-term reactivity by using a moderate action horizon (*e.g.*, half of the prediction length), which is inevitably sup-optimal when both factors are prominent. Temporal ensembling strengthens dependency across chunks by averaging multiple decisions over time; however, weighted-averaging operations can be detrimental when consecutive decisions fall into distinct modes. Our method more effectively addresses cross-chunk dependency through dedicated behavioral search and is not mutually exclusive to previous methods. We will demonstrate in the next section that combining our method with the moving average can further improve closed-loop action chunking.

| name | value |
|---|---|
| batch size $N$ | 16 |
| mode size $K$ | 3 |
| prediction length $l$ | 16 |
| temporal coherence decay $\rho$ | 0.5 |
| moving average decay $\lambda$ | 0.5 |

Table 6: Default hyper-parameters in our experiments.

## C  ADDITIONAL DETAILS

### C.1  SIMULATION EXPERIMENT DETAILS

#### C.1.1  ENVIRONMENT DETAILS

Our simulation experiments are conducted on three robot manipulation benchmarks. We use the training data collected from human demonstrations in each benchmark.

*Push-T*: We adopt the Push-T environment introduced in (Chi et al., 2023), where the goal is to push a T-shaped block on a table to a target position. The action space is two-dimensional end-effector velocity control. The training dataset contains 206 demonstrations collected by humans.

*Robomimic*: We use five tasks in the Robomimic suite (Mandlekar et al., 2022), namely Lift, Can, Square, Transport, and Tool Hang. The training dataset for each task contains 300 episodes collected from multi-human (MH) demonstrations.

*Franka Kitchen*: We use the Franka Kitchen environment from (Gupta et al., 2020), featuring a Franka Panda arm with a seven-dimensional action space and 566 human-collected demonstrations. The learned policy is evaluated on test cases involving four or more objects (p4), a challenging yet practical task for robotic manipulation in household contexts.

#### C.1.2  IMPLEMENTATION DETAILS.

Our implementation of BID for Diffusion Policy is built upon the official code of Chi et al. (2023), with modifications made solely to the inference process. The policy takes in state inputs and predicts a chunk of 16 actions as outputs. For each simulation task, we train the model for 100-1000 epochs to reach near-optimal performance. We evaluate it in closed-loop operations, *i.e.*, action horizon is set to 1. For forward contrast, we train the weak policy for 10-100 epochs, resulting in a suboptimal policy for each task. The core hyperparameters are summarized in Table 6.

Our implementation of BID for VQ-BeT (Lee et al., 2024) is built upon the code of LeRobot (Cadene et al., 2024). We use the best public checkpoint as the strong policy and a checkpoint trained for 100k iterations as the weak policy. Since BID requires sample diversity, we set the temperature to be 0.5 for all methods in our experiments.

### C.2  REAL-WORLD EXPERIMENT DETAILS

#### C.2.1  DYNAMIC PLACING

**Task.**  We consider a task where the robot is to deliver an object held in its gripper into a cup held by a human. As shown in Fig. 14, this task comprises four main stages and presents two core challenges. First, due to the similar size of the object and the cup, the robot must achieve high precision to place the object accurately into the cup. Second, the position of the cup is not fixed, requiring the robot to adjust its plans based on the latest position continuously. This task mirrors real-world scenarios where robots interact with a dynamic environment, accommodating moving objects and agents.

**Demonstration.**  In light of temporal dependencies and style variations in human behaviors, we intentionally collect a diverse set of demonstration data, differing in factors such as average speed, idling pause, and overall trajectory. We gather a total of 150 demonstration episodes: 50 clean and consistent demonstrations, and 100 noisy and diverse demonstrations. All demonstrations successfully accomplish the task. Additional, the location of the cup is fixed and static within each episode.

**Robot.**  Following previous works (Chi et al., 2023; Prasad et al., 2024), we use a Franka Panda as the robot hardware and the vision-based diffusion policy for its operation. The robot is equipped

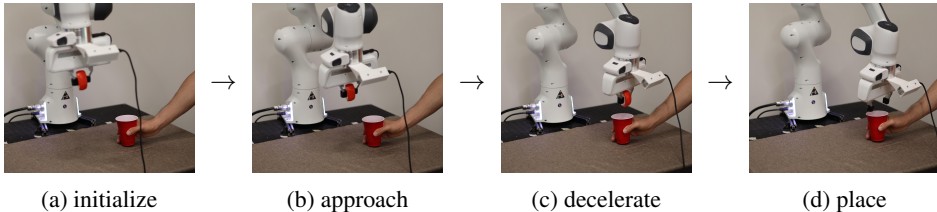

| (a) initialize | (b) approach | (c) decelerate | (d) place |

Figure 14: Human demonstrations on a Franka Panda robot for a real-world object delivery task. The robot is tasked with delivering an object held in its gripper into a cup held by a human. Each demonstration consists of four main stages: (a) initialize the robot position randomly, (b) approach the target cup, (c) slow down near the target cup, and (d) release the object. The position of the target cup may change during an episode.

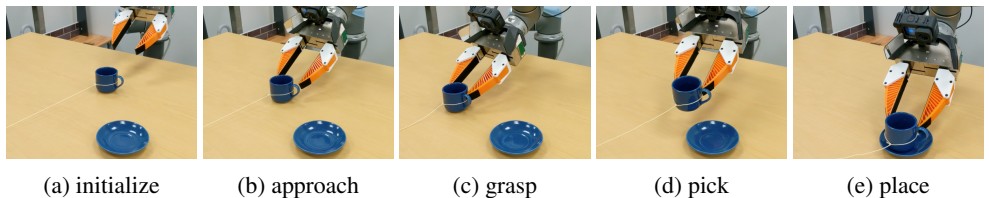

| (a) initialize | (b) approach | (c) grasp | (d) pick | (e) place |

Figure 15: The robot is tasked with picking up a cup and placing it on a saucer nearby. The four main stages are (a) initializing the robot, (b) approaching the target cup, (c) grasping the target cup, (d) picking up the cup, and (e) placing the cup on the target saucer. The position of the target cup may change during an episode.

with two cameras: one ego-centric camera mounted at the wrist of the robot, one third-person camera mounted at a static bracket. Both cameras provide visual observations at a resolution of $256 \times 256$ pixels. The robot operates at a frequency of 10 Hz, with a prediction length of 16 time steps.

**Evaluation.** We evaluate our method in comparison to vanilla random sampling under two conditions: *static target*, where the target cup remains fixed throughout the evaluation, and *dynamic target*, where the target cup is gradually moved. In the dynamic setting, the location of the cup stays within the range of training locations, but the movement is not encountered during training. This evaluation protocol is designed to explicitly assess the ability of the policy to react to unexpected dynamics in the environment. Each method-setting pair is tested over 20 episodes, with both the initial and target locations randomized across different episodes.

**Result.** We summarize the result of the real-world experiments in Fig. 8. The success rate of vanilla random sampling is generally limited due to oscillations between different latent strategies, which quickly diverge from the distribution of demonstrations. This issue is particularly pronounced in the dynamic setting, where the vanilla baseline struggles to account for the target movements within an action chunk lasting for 1.6 seconds. In contrast, the proposed BID method significantly improves performance in both static and dynamic settings. Notably, BID maintains a similar success rate in the dynamic setting as in the static setting, suggesting its potential to extend action chunking into uncertain environments.

### C.2.2 DYNAMIC PICKING

**Task.** Next, we consider a task where the robot is required to pick up a cup and place it onto a nearby saucer. The cup was pulled with a string until the robot's gripper successfully grasped it. The task consists of five main stages, which are illustrated in Fig. 15. This setup also tests the robot's capability to interact with a dynamic environment, a critical challenge in real-world applications.

**Policy.** We utilized the publicly available diffusion policy checkpoint from UMI (Chi et al., 2024) without any additional fine-tuning. Notably, the policy was originally trained using demonstrations in a static setting, where the cup's position remained constant throughout the task. Our experimental setup mirrored the one described by UMI, using the same UR5 robot hardware. This allowed us to directly evaluate the policy's transferability to a dynamic environment, where the cup's position changes during the task. Due to the absence of an early checkpoint, we omitted negative samples in forward contrast, focusing solely on positive consistency discussed in Appendix A.3.

**Evaluation.** We evaluated BID against three baselines: vanilla random sampling in both open-loop and closed-loop configurations, and EMA (closed-loop). These methods were tested under two conditions: *static target*, where the cup remained in a fixed position, and *dynamic target*, where the cup was moved using the string. Each method-setting combination was tested across 20 episodes, with the initial positions of the cup and saucer kept consistent to ensure controlled comparisons.

**Results.** The results, summarized in Fig. 9, highlight the challenges of the dynamic setting. Open-loop vanilla sampling performed poorly due to its inability to adapt to the cup's movements, often failing to approach the cup as it was pulled. While closed-loop vanilla sampling showed improved reactivity, it suffered from inconsistent trajectories, resulting in jittery behavior when attempting to grasp and place the cup. Similarly, closed-loop EMA sampling demonstrated higher adaptability to environmental changes but often failed to firmly grasp the cup, likely due to the limitations of naive averaging, which compromises commitment to a specific strategy. In contrast, BID achieved at least a 2x improvement in success rate compared to all other methods in the dynamic setting, while maintaining its performance in the static setting, demonstrating both adaptability and precision in dynamic environments.

## D    ADDITIONAL DISCUSSIONS

**Relation to Option Discovery.** Action chunking and option discovery share similarities in modeling temporally extended actions. However, their designs and outcomes are often different. Option discovery typically aims to learn hierarchical policies, explicitly discovering high-level skills from low-level action sequences (Bagaria & Konidaris, 2020; Jiang et al., 2022; 2023; Zhao et al., 2023a). In contrast, action chunking operates directly on low-level action sequences with fixed horizons, bypassing the need for abstraction into high-level skills. This simplicity has proven effective in large-scale robotic foundation models (Team et al., 2024; Black et al., 2024), addressing the scalability challenge that option discovery has yet to overcome. Future work could explore variable-horizon option discovery as a compelling alternative to action chunking, potentially combining the benefits of temporal abstraction with the scalability demonstrated by action chunking.

**Relation to Long Context.** Our analysis in §3 highlights the benefits of extending the action horizon to better capture temporal dependencies across actions. Another natural approach to capturing these dependencies is to extend the context length. However, long-context policies often suffer from robustness issues, due to spurious correlations between past and future actions, as evidenced in Appendix A2 and studied in de Haan et al. (2019); Wen et al. (2020). Nevertheless, the increasing availability of large-scale robotic datasets for pre-training and fine-tuning may help mitigate these challenges. Advances in leveraging long context may open up new opportunities in policy design, such as long-context transformers (Su et al., 2024) and recurrent neural networks (Hochreiter & Schmidhuber, 1997; Zhuo et al., 2020).

## E    PROOFS

In this section, we will clearly write out the transition dynamics in the training environment to be $P_{\text{train}}(s_{t+1} \mid s_t, a_t)$, the transition dynamics in the test environment to be $P_{\text{test}}(s_{t+1} \mid s_t, a_t)$ and the implicit dynamics model of the training environment to be $\hat{P}_{\text{train}}(s_{t+1} \mid s_t, a_t)$. Note that the implicit dynamics model attempts to learn the dynamics of the training environment and not the test environment since our policy is assumed to have been trained only on data from the training environment.

First, we establish the following lemma which will help us compare different function classes based on the information they have access to:

**Lemma 4.** *Let $\mathcal{L}$ be a convex function and let $X$ and $Y$ be two random variables. Let $G$ be the class of functions $g(X)$ that accept $X$ as an input. Then*

$$\min_{g(X) \in G} \mathbb{E}_{X,Y} \left[ \mathcal{L}(f(X,Y), g(X)) \right] = \mathbb{E}_X \left[ \min_{c \in \mathbb{R}} \mathbb{E}_Y \left[ \mathcal{L}(f(X,Y), c) | X \right] \right].$$

*Proof.* The left hand side is less than or equal to the right hand side by the following logic:

$$\mathbb{E}_X \left[ \min_{c \in \mathbb{R}} \mathbb{E}_Y \left[ \mathcal{L}(f(X,Y), c) | X \right] \right] = \mathbb{E}_X \left[ \mathbb{E}_Y \left[ \mathcal{L}(f(X,Y), c^*(X)) | X \right] \right]$$
$$\geq \min_{g(X) \in G} \mathbb{E}_{X,Y} \left[ \mathcal{L}(f(X,Y), g(X)) \right]$$

where we used $c^*(X) := \arg\min_c \mathbb{E}_X[\mathcal{L}(f(X,Y), c) | X]$. We get the inequality by recognizing that $\mathbb{R} \subsetneq G$. For the reverse inequality, consider any $g(X) \in G$:

$$\mathbb{E}[\mathcal{L}(f(X,Y), g(X))] = \mathbb{E}_X \left[ \mathbb{E}_Y \left[ \mathcal{L}(f(X,Y), g(X)) | X \right] \right]$$
$$\geq \mathbb{E}_X \left[ \min_g \mathbb{E}_Y \left[ \mathcal{L}(f(X,Y), g(X)) | X \right] \right]$$
$$= \mathbb{E}_X \left[ \min_c \mathbb{E}_Y \left[ \mathcal{L}(f(X,Y), c) | X \right] \right].$$

With these two inequalities, we conclude. $\qquad\square$

Next, we prove the following lemma. This straightforward, and almost trivial, result is provided as a separate lemma because we simplify terms in this manner quite often throughout our proofs.

**Lemma 5.** *Let $\mathcal{L}$ be a convex function and let $X, Y$ be two random variables. Then,*

$$\min_f \mathbb{E}_{X,Y} \left[ P(X' = X)\mathcal{L}(f(X'), S(X,Y)) \right] + \mathbb{E}_{X,Y} \left[ \sum_{X' \neq X} P(X')\mathcal{L}\left(f(X'), S(X,Y)\right) \right]$$
$$\leq \min_f \{ \mathbb{E}_{X,Y} \left[ \mathcal{L}(f(X), S(X,Y)) \right] \} + \epsilon$$

*where $\epsilon = \max_{X' \neq X, X, Y} \{ \mathcal{L}(f^*(X'), S(X,Y) \}$ and $f^* = \arg\min_f \{ \mathbb{E}_{X,Y} \left[ \mathcal{L}(f(X), S(X,Y)) \right] \}$.*

*Proof.*

$$\min_f \mathbb{E}_{X,Y} \left[ P(X' = X)\mathcal{L}(f(X'), S(X,Y)) \right] + \mathbb{E}_{X,Y} \left[ \sum_{X' \neq X} P(X')\mathcal{L}\left(f(X'), S(X,Y)\right) \right]$$

$$\leq \min_f \mathbb{E}_{X,Y} \left[ \mathcal{L}(f(X), S(X,Y)) \right] + \mathbb{E}_{X,Y} \left[ \sum_{X' \neq X} P(X')\mathcal{L}\left(f(X'), S(X,Y)\right) \right]$$

$$\leq \min_f \{ \mathbb{E}_{X,Y} \left[ \mathcal{L}(f(X), S(X,Y)) \right] \} + \mathbb{E}_{X,Y} \left[ \sum_{X' \neq X} P(X')\mathcal{L}\left(f^*(X'), S(X,Y)\right) \right]$$

$$\leq \min_f \{ \mathbb{E}_{X,Y} \left[ \mathcal{L}(f(X), S(X,Y)) \right] \} + \mathbb{E}_{X,Y} \left[ \sum_{X' \neq X} P(X')\epsilon \right]$$

$$\leq \min_f \{ \mathbb{E}_{X,Y} \left[ \mathcal{L}(f(X), S(X,Y)) \right] \} + \epsilon$$

$\qquad\square$

## E.1 ASSUMPTIONS

Considering that recent policies often use a short context length $c$, we assume the range of temporal dependency modeled by a $(c, h)$-policy is limited:

**Assumption 1.** The sum of context length and action horizon is less than the length of temporal dependency in expert demonstrations, $c + h < k$.

This assumption allows us to focus on the problem that is relevant to us and allows us to ignore edge cases. However, our analysis *can* be extended to the case where this assumption does not hold. In the case where $c + h \geq k$, the larger action chunk model will not get any advantage (making $\alpha_b = 0$) since the additional states it has observed in time steps $\tau_b$ are irrelevant to $a_t$. Then, the longer horizon policy either attains the same expected loss the shorter horizon policy or suffer greater loss from not having observed the recent past states.

**Assumption 2.** An optimal $\pi_{c,h}$ must infer the unobserved states based on the observed states and actions by modeling the transition dynamics $\hat{P}_{\text{train}}(s_{t'} \mid s_{t'-1}, a_{t'-1})$ accurately for all time step $t'$. We consider this implicit model to be the same for both $\pi_{(c,h)}$ and $\pi_{(c,h+d)}$.

In other words, we can write using the law of total probability:

$$
\pi_{(c,h)}(a_t | s_{t-h-c:t-h}, a_{t-h:t-1})
$$
$$
= \mathbb{E}_{s_{t-k:t-h-c-1}, s_{t-h+1:t} \sim \hat{P}_{\text{train}}} \left[ \pi_{(k,1)}(a_t | s_{t-k:t}) | s_{t-h-c:t-h}, a_{t-h:t-1} \right]
$$

and

$$
\pi_{(c,h+d)}(a_t | s_{t-h-d-c:t-h-d}, a_{t-h-d:t-h-1})
$$
$$
= \mathbb{E}_{s_{t-k:t-h-d-c-1}, s_{t-h-d+1:t} \sim \hat{P}_{\text{train}}} \left[ \pi_{(k,1)}(a_t | s_{t-k:t}) | s_{t-h-d-c:t-h-d}, a_{t-h-d:t-1} \right].
$$

## E.2 DEFINITIONS

We, first, analyze the effect of reducing context horizon. We show that, provided action horizon is constant, decreasing context horizon causes performance of the optimal policy to decrease.

Consider a $(c, h)$-policy (i.e., the policy has context length $c$ and action horizon $h$) whose probability of taking action $a_t$ at time $t$ in a chunk generated at $t$ is referred to as $\pi_{(c,h)} := \pi_{(c,h)}(a_t | s_{t-c:t})$. On the other hand, consider a $(c+1, h)$-policy whose probability of taking action $a_t$ in a chunk generated at time $t$ is referred to as $\pi_{(c+1,h)} := \pi_{(c+1,h)}(a_t | s_{t-c-1:t})$. Lastly, consider a $(k, 1)$-expert whose probability of taking action $a_t$ at time $t$ is $\pi^*$.

**Proposition 6.** *Let $\mathcal{L}$ be a non-linear, convex function. Let $c < k$. Let $G := \{a_t, s_{t-k:t-c-1}, z_{t-k:t}\}$ and let $C := \{s_{t-c:t}\}$. Then,*

$$
\min_{\pi_{(c+1,h)}} \mathbb{E}_G \left[ \mathcal{L}(\pi_{(c+1,h)}, \pi^*) \Big| C \right] \leq \min_{\pi_{(c,h)}} \mathbb{E}_G \left[ \mathcal{L}(\pi_{(c,h)}, \pi^*) \Big| C \right]
$$

*In particular, this is an equality if and only if $s_{t-c-1} \sim P_{\text{test}}(\cdot \mid C)$ and $\hat{s}_{t-c-1} \sim \hat{P}_{\text{train}}(\cdot \mid C)$ take on only one and the same value almost surely.*

*Proof.* We refer to the class of functions that accept $a_t$ and $s_{t-c-1:t}$ as inputs as $X_2$. Similarly, the class of functions that do not accept $a_t$ as inputs but accept $s_{t-c-1:t}$ as inputs is $X_1$. The function class that accepts only $s_{t-c:t}$ and not $s_{t-c-1}$ or $a_t$ as inputs are elements of $X_0$. Lastly, the function class that accepts $s_{t-c:t}$ and $a_t$ as inputs, but not $s_{t-c-1}$, are elements of $X_{-1}$.

$$
\min_{\pi_{(c+1,h)} \in X_2} \mathbb{E}_G \left[ \mathcal{L}(\pi_{(c+1,h)}, \pi^* \Big| C \right]
$$

$$
= \mathbb{E}_{a_t} \left[ \min_{\pi'_{(c+1,h)} \in X_1} \mathbb{E}_{s_{t-c-1}} \left[ \mathbb{E}_{s_{t-k:t-c-2}, z_{t-k:t}} \left[ \mathcal{L}(\pi'_{(c+1,h)}, \pi^*) \Big| a_t, s_{t-c-1}, C \right] \Big| a_t, C \right] \Big| C \right]
$$
(Lemma 4)

$$
= \mathbb{E}_{a_t} \left[ \mathbb{E}_{s_{t-c-1}} \left[ \min_{\pi'_{(c,h)} \in X_0} \mathbb{E}_{s_{t-k:t-c-2}, z_{t-k:t}} \left[ \mathcal{L}(\pi'_{(c,h)}, \pi^*) \Big| a_t, s_{t-c-1}, C \right] \Big| a_t, C \right] \Big| C \right] \quad \text{(Lemma 4)}
$$

$$
\leq \mathbb{E}_{a_t} \left[ \min_{\pi'_{(c,h)} \in X_0} \mathbb{E}_{s_{t-c-1}} \left[ \mathbb{E}_{s_{t-k:t-c-2}, z_{t-k:t}} \left[ \mathcal{L}(\pi'_{(c,h)}, \pi^*) \Big| a_t, s_{t-c-1}, C \right] \Big| a_t, C \right] \Big| C \right]
$$
(Jensen's inequality)

$$= \min_{\pi_{(c,h)} \in X_{-1}} \mathbb{E}_{a_t} \left[ \mathbb{E}_{s_{t-c-1}} \left[ \mathbb{E}_{s_{t-k:t-c-2}, z_{t-k:t}} \left[ \mathcal{L}(\pi_{(c,h)}, \pi^*) \Big| a_t, s_{t-c-1}, C \right] \Big| a_t, C \right] \Big| C \right] \quad \text{(Lemma 4)}.$$

Use the law of total expectation to conclude. The equality conditions for Jensen's inequality provides us the equality condition for this relationship too. □

Now, we formalize the definitions of *Expected Observation Advantage* and *Maximum Inference Disadvantage*.

Recall that, in §3.2, we have two policies: $\pi_{(c,h)}$ and $\pi_{(c,h+d)}$; the former sees more recent states while the latter remembers more past states. First, we define an agent that gets access to all the information that both learners, combined, have: a $(c + d, h)$-policy whose probability of taking action $a_t$ in a chunk generated at time $t - h$ is

$$\pi_{(c+d,h)} := \pi_{(c+d,h)}(a_t | s_{t-h-d-c:t-h}, a_{t-h:t-1}).$$

Observe that $\pi_{(c+d,h)}$ has access to more context than $\pi_{(c,h)}$, particularly the knowledge of states $s_{t-h-c-d:t-h-c-1}$.

**Definition (Expected Observation Advantage ($\alpha_b$)).** We know, using Proposition 6, $\pi_{(c+d,h)}$ has lower divergence with respect to $\pi^*$ than $\pi_{(c,h)}$. We say that the advantage $\pi_{(c+d,h)}$ gets from the extra information is $\alpha_b$. More formally, we say that

$$\alpha_b := \min_{\pi_{(c,h)}} \mathbb{E}\left[ \mathcal{L}(\pi_{(c,h)}, \pi^*) \Big| C \right] - \min_{\pi_{(c+d,h)}} \mathbb{E}\left[ \mathcal{L}(\pi_{(c+d,h)}, \pi^*) \Big| C \right] \tag{8}$$

where $C$ is defined as in Proposition 1. Clearly, $\alpha_b \geq 0$. In particular, $\alpha_b = 0$ when $s_{t-h-d-c:t-h-c-1}$ can be deterministically inferred by $\pi_{(c,h)}$ or when the expert policy is independent of them.

**Definition (Maximum Inference Disadvantage ($\epsilon_f$)).** Consider the maximum divergence that can be accumulated by the $(c, h + d)$-policy from not knowing the recent states at time steps $s_{t-h-d+1:t-h}$, and let that be $\epsilon_f$. More formally, we say that, for fixed $C$ from Proposition 1, any states in $\mathcal{S}^- := \{s_{t-k:t}\} \setminus \mathcal{S}^+$, any $z_{t-k:t}$, and any $\hat{s}_{t-h-d+1:t-h} \neq s_{t-h-d+1:t-h}$, the following holds:

$$\mathcal{L}(\pi_{(c+d,h)}(a_t | s_{t-h-d-c:t-h-d}, \hat{s}_{t-h-d+1:t-h} \neq s_{t-h-d+1:t-h}, a_{t-h:t-1}), \pi^*) \leq \epsilon_f. \tag{9}$$

Here, $\pi_{(c+d,h)} := \arg\min_{\pi_{(c+d,h)}} \mathbb{E}[\mathcal{L}(\pi_{(c+d,h)}, \pi^*)|C]$ is the optimal $(c + d, h)$-policy.

To define $\alpha_f$ and $\epsilon_b$, we prove a second version of Proposition 6. Consider a $(c, h)$-policy whose probability of taking action $a_t$ at time $t$ in a chunk generated at $t$ is referred to as $\pi_{(c,h)} := \pi_{(c,h)}(a_t | s_{t-c:t})$. On the other hand, consider a $(c - 1, h + 1)$-policy whose probability of taking action $a_t$ in a chunk generated at time $t - 1$ is referred to as $\pi_{(c-1,h+1)} := \pi_{(c-1,h+1)}(a_t | s_{t-c:t-1})$. Lastly, consider a $(k, 1)$-expert whose probability of taking action $a_t$ at time $t$ is $\pi^*$.

**Proposition 7.** *Let $\mathcal{L}$ be a non-linear, convex function. Let $c < k$. Let $G := \{a_t, s_{t-k:t-c-1}, s_t, z_{t-k:t}\}$ and $C := \{s_{t-c:t-1}, a_{t-1}\}$. Then,*

$$\min_{\pi_{(c,h)}} \mathbb{E}_G \left[ \mathcal{L}(\pi_{(c,h)}, \pi^*) \Big| C \right] \leq \min_{\pi_{(c-1,h+1)}} \mathbb{E}_G \left[ \mathcal{L}(\pi_{(c-1,h+1)}, \pi^*) \Big| C \right].$$

*In particular, this is an equality if and only if the state $s_t \sim P_{\text{test}}(\cdot \mid C)$ and $\hat{s}_t \sim \hat{P}_{\text{train}}(\cdot \mid C)$ takes on only one and the same value almost surely.*

*Proof.* The proof is similar to that of Proposition 6. We refer to the class of functions that accept $a_t$ and $s_{t-c:t}$ as inputs as $X_2$. Similarly, the class of functions that do not accept $a_t$ as inputs but accept $s_{t-c:t}$ as inputs is $X_1$. The function class that accepts only $s_{t-c:t-1}$ and not $s_t$ or $a_t$ as inputs

are elements of $X_0$. Lastly, the function class that accepts $s_{t-c:t-1}$ and $a_t$ as inputs, but not $s_t$, are elements of $X_{-1}$.

$$\min_{\pi_{(c,h)} \in X_2} \mathbb{E}_G \left[ \mathcal{L}(\pi_{(c,h)}, \pi^* \big| C \right]$$

$$= \mathbb{E}_{a_t} \left[ \min_{\pi'_{(c,h)} \in X_1} \mathbb{E}_{s_t} \left[ \mathbb{E}_{s_{t-k:t-c-1}, z_{t-k:t}} \left[ \mathcal{L}(\pi'_{(c,h)}, \pi^*) \big| a_t, s_t, C \right] \big| a_t, C \right] \big| C \right] \quad \text{(Lemma 4)}$$

$$= \mathbb{E}_{a_t} \left[ \mathbb{E}_{s_t} \left[ \min_{\pi'_{(c-1,h+1)} \in X_0} \mathbb{E}_{s_{t-k:t-c-2}, z_{t-k:t}} \left[ \mathcal{L}(\pi'_{(c-1,h+1)}, \pi^*) \big| a_t, s_t, C \right] \big| a_t, C \right] \big| C \right] \quad \text{(Lemma 4)}$$

$$\leq \mathbb{E}_{a_t} \left[ \min_{\pi'_{(c-1,h+1)} \in X_0} \mathbb{E}_{s_t} \left[ \mathbb{E}_{s_{t-k:t-c-1}, z_{t-k:t}} \left[ \mathcal{L}(\pi'_{(c-1,h+1)}, \pi^*) \big| a_t, s_t, C \right] \big| a_t, C \right] \big| C \right]$$
$$\text{(Jensen's inequality)}$$

$$= \min_{\pi_{(c-1,h+1)} \in X_{-1}} \mathbb{E}_{a_t} \left[ \mathbb{E}_{s_t} \left[ \mathbb{E}_{s_{t-k:t-c-1}, z_{t-k:t}} \left[ \mathcal{L}(\pi_{(c-1,h+1)}, \pi^*) \big| a_t, s_t, C \right] \big| a_t, C \right] \big| C \right] \quad \text{(Lemma 4)}.$$

Use the law of total expectation to conclude. The equality condition can be seen from the equality condition of Jensen's inequality. $\qquad \square$

Using this, we can define $\epsilon_b$ and $\alpha_f$ in a similar manner:

**Definition (Expected Observation Advantage ($\alpha_f$)).** Recall that we have two models: $\pi_{(c,h)}$ and $\pi_{(c,h+d)}$ and a hypothetical $(c+d,h)$-policy that has access to all the information both our learners have (as in Eq. (8) and Eq. (9)). Observe that $\pi_{(c+d,h)}$ has access to more context than $\pi_{(c,h+d)}$, particularly the knowledge of states $s_{t-h-d+1:t-h}$. Therefore, we know, using Proposition 7, $\pi_{(c+d,h)}$ has lower divergence with respect to $\pi^*$ than $\pi_{(c,h+d)}$. We say that the advantage $\pi_{(c+d,h)}$ gets from the extra information is $\alpha_f$. More formally, we say that

$$\alpha_f = \min_{\pi_{(c,h+d)}} \mathbb{E}\left[\mathcal{L}(\pi_{(c,h+d)}, \pi^*) \big| C\right] - \min_{\pi_{(c+d,h)}} \mathbb{E}\left[\mathcal{L}(\pi_{(c+d,h)}, \pi^*) \big| C\right] \qquad (10)$$

where $C$ is defined as in Proposition 1. Clearly, $\alpha_f \geq 0$. In particular, $\alpha_f = 0$ when $\pi_{(c,h+d)}$ can infer $s_{t-h-d+1:t-h}$ perfectly. This makes sense–in the static environment, observing these states does not provide any advantage since the optimal $\pi_{(c,h+d)}$ can infer these states anyway using the actions taken at those time steps (assuming that the implicit dynamics model is accurate).

**Definition (Maximum Inference Disadvantage ($\epsilon_b$)).** Consider the maximum divergence that can be accumulated by the $(c,h)$-model from not knowing the past states $s_{t-h-d-c:t-h-c-1}$ and let that be $\epsilon_b$. More formally, we say that, for fixed $C$ from Proposition 1, any states in $\mathcal{S}^-$, any $z_{t-k:t}$ and any $\hat{s}_{t-h-d-c:t-h-c-1} \neq s_{t-h-d-c:t-h-c-1}$:

$$\mathcal{L}(\pi_{(c+d,h)}(a_t | \hat{s}_{t-h-d-c:t-h-c-1} \neq s_{t-h-d-c:t-h-c-1}, s_{t-h-c:t-h}, a_{t-h:t-1}), \pi^*) \leq \epsilon_b. \quad (11)$$

Here, $\pi_{(c+d,h)} := \arg\min_{\pi_{(c+d,h)}} \mathbb{E}[\mathcal{L}(\pi_{(c+d,h)}, \pi^*) | C]$ is the optimal $(c+d,h)$-policy.

As a warmup, we see that the intuitive relationship between $\alpha_f$ and $\epsilon_f$ (and the same for $\alpha_b$ and $\epsilon_b$) holds:

**Proposition 8.** $\alpha_f \leq \epsilon_f$ and $\alpha_b \leq \epsilon_b$.

*Proof.* We prove the first inequality; the second can be proven in the same manner. We use Assumption 2 to write $\pi_{(c,h+d)} = \mathbb{E}_{s_{t-h-d+1:t-h} \sim P} \left[\pi_{(c+d,h)} \mid s_{t-h-d-c:t-h-d}, a_{t-h-d:t-1}\right]$ where $P$ is the environment's transition dynamics. Let

$$P_{\text{correct inference}} = P(\hat{s}_{t-h-d+1:t-h} = s_{t-h-d+1:t-h} | s_{t-h-d-c:t-h-d}, a_{t-h-d:t-1})$$

and

$$P_{\text{incorrect inference}} = P(\hat{s}_{t-h-d+1:t-h} \neq s_{t-h-d+1:t-h} | s_{t-h-d-c:t-h-d}, a_{t-h-d:t-1}).$$

Then,

$$
\begin{aligned}
\alpha_f &= \min_{\pi_{(c,h+d)}} \mathbb{E}\left[\mathcal{L}(\pi_{(c,h+d)}, \pi^*)\Big|C\right] - \min_{\pi_{(c+d,h)}} \mathbb{E}\left[\mathcal{L}(\pi_{(c+d,h)}, \pi^*)\Big|C\right] \\
&= \min_{\pi_{(c+d,h)}} \mathbb{E}\left[\mathcal{L}(P_{\text{correct inference}}\pi_{(c+d,h)} + P_{\text{incorrect inference}}\pi_{(c+d,h)}, \pi^*)\Big|C\right] \\
&\quad - \min_{\pi_{(c+d,h)}} \mathbb{E}\left[\mathcal{L}(\pi_{(c+d,h)}, \pi^*)\Big|C\right] \\
&\leq \min_{\pi_{(c+d,h)}} \Big\{\mathbb{E}\left[P_{\text{incorrect inference}}\mathcal{L}(\pi_{(c+d,h)}(\text{conditioning on incorrect inference}), \pi^*)\Big|C\right] \\
&\quad + \mathbb{E}\left[P_{\text{correct inference}}\mathcal{L}(\pi_{(c+d,h)}(\text{conditioning on correct inference}), \pi^*)\Big|C\right]\Big\} \\
&\quad - \min_{\pi_{(c+d,h)}} \mathbb{E}\left[\mathcal{L}(\pi_{(c+d,h)}, \pi^*)\Big|C\right] \hspace{3cm} \text{(Convexity)} \\
&\leq \mathbb{E}\left[P_{\text{incorrect inference}}\mathcal{L}(\hat{\pi}^*_{(c+d,h}(\text{conditioning on incorrect inference}), \pi^*)\Big|C\right] \\
&\quad + \mathbb{E}\left[\mathcal{L}(\pi^*_{(c+d,h)}, \pi^*)\Big|C\right] - \mathbb{E}\left[\mathcal{L}(\pi^*_{(c+d,h)}, \pi^*)\Big|C\right] \\
&\hspace{4cm} \text{(Bounding probabilities by 1 and Lemma 5)} \\
&\leq \mathbb{E}\left[P_{\text{incorrect inference}}\epsilon_f \mid C\right] \\
&\leq \epsilon_f
\end{aligned}
$$

Here, $\pi^*_{(c+d,h)} := \arg\min_{\pi_{(c+d,h)}} \mathbb{E}\left[\mathcal{L}(\pi_{(c+d,h)}, \pi^*)\Big|C\right]$. $\hspace{2cm}$ □

We will provide a tighter bound after the proof of our main theoretical results.

**Definition (Forward and Backward Inference).** For a fixed time step $t$ and $C$ (as in §3.2), consider the time steps $\tau_f := \{t - h - d + 1 : t - h\}$ and $\tau_b := \{t - h - d - c : t - h - c - 1\}$.
Define
$$
P_f(t') := P_{\text{test}}(S_{t'} = g_{t'} \mid S_{t'-1} = g_{t'-1}, A_{t'-1} = a_{t'-1})
$$
for any $t' \in \tau_f$ with $g_{t'}, g_{t'-1}, a_{t'-1}$ being the ground truth states and action in the deterministic test environment. As such, $P_f(t') = 1$ in a deterministic environment. In a stochastic environment, $P_f(t') < 1$ for all $t'$ and as the stochasticity increases, these values decrease and approach 0. Let
$$
\delta_f(t') = \hat{P}_{\text{train}}(g_{t'}|g_{t'-1}, a_{t'-1}) - P_f(t')
$$
where $g_{t'}, g_{t'-1}, a_{t'-1}$ are still the ground truth in the deterministic test environment. Define $\hat{P}_f(t') = P_f(t') + \delta_f(t')$. Similarly, define
$$
P_b(t') := P_{\text{test}}(S_{t'} = g_{t'} \mid S_{t'+1} = g_{t'+1})
$$
for any $t' \in \tau_b$ where $g_{t'}, g_{t'-1}$ are the ground truth states in the deterministic test environment. Then let
$$
\delta_b(t') = \hat{P}_{\text{train}}(g_{t'}|g_{t'+1}) - P_b(t')
$$
for any $t' \in \tau_b$. Define $\hat{P}_b(t') = P_b(t') + \delta_b(t')$.

Intuitively, $\delta_f(t')$ and $\delta_b(t')$ characterize the error that the implicit dynamics model has in capturing the test environment's dynamics.

### E.3 CONSISTENCY-REACTIVITY INEQUALITIES

Now we prove the Consistency-Reactivity Inequalities.

**Proposition 1** (Consistency-Reactivity Inequalities) Let $\mathcal{L}$ be a non-linear and non-negative convex function measuring the prediction error with respect to demonstrations. Let $\mathcal{S}^+ \subset \{s_{t-k:t}\}$ be the states both the $(c, h)$ and the $(c, h + d)$ policies observe and let $\mathcal{S}^- := \{s_{t-k:t}\} \setminus \mathcal{S}^+$. Let $C := \{a_{t-h:t-1}\} \cup \mathcal{S}^+$, $G := \{a_t, z_{t-k:t}\} \cup \{a_{t-h-d:t-h-1}\} \cup \mathcal{S}^-$. For notational ease, let $\tau_f =$

$\{t - h - d + 1 : t - h\}$ and let $\tau_b = \{t - h - d - c : t - h - c - 1\}$. Then, we can bound the expected loss of the $(c, h + d)$-policy and the $(c, h)$-policy as:

$$\alpha_f - \epsilon_b(1 - \prod_{t' \in \tau_b} P_b(t')(P_b(t') + \delta_b(t'))) \leq \min_{\pi_{h+d}} \mathbb{E}_G\left[\mathcal{L}(\pi_{h+d}, \pi^*)|C\right] - \min_{\pi_h} \mathbb{E}_G\left[\mathcal{L}(\pi_h, \pi^*)|C\right]$$

and

$$\min_{\pi_{h+d}} \mathbb{E}_G\left[\mathcal{L}(\pi_{h+d}, \pi^*)|C\right] - \min_{\pi_h} \mathbb{E}_G\left[\mathcal{L}(\pi_h, \pi^*)|C\right] \leq \epsilon_f(1 - \prod_{t' \in \tau_f} P_f(t')(P_f(t') + \delta_f(t'))) - \alpha_b.$$

*Proof.* We first prove the upper bound. For ease of notation, we will write $x_{a:}^b$ to mean $x_{a:b}$. Additionally, for greater clarity, we will explicitly include the context length of each model, so $\pi_{(c,h)} = \pi_h$ and $\pi_{(c,h+d)} = \pi_{h+d}$. We start by writing, using Assumption 2,

$$\pi_{(c,h+d)}(a_t | s_{t-h-d-c:t-h-d}, a_{t-h-d:t-1})$$
$$= \pi_{(c,h+d)}(a_t | s_{t-h-d-c:}^{t-h-d}, a_{t-h-d:}^{t-1})$$
$$= \mathbb{E}_{\hat{s}_{t-h-d+1:t-h}}\left[\pi_{(c+d,h)}(a_t | s_{t-h-d-c:}^{t-h-d}, \hat{s}_{t-h-d+1:}^{t-h}, a_{t-h-d:}^{t-1}) \Big| s_{t-h-d-c:}^{t-h-d}, a_{t-h-d:}^{t-1}\right].$$

Using this, we expand the left hand side of our inequality:

$$\min_{\pi_{(c,h+d)}} \mathbb{E}_G\left[\mathcal{L}(\pi_{(c,h+d)}, \pi^*)|C\right]$$
$$= \min_{\pi_{(c+d,h)}} \mathbb{E}_G\left[\mathcal{L}(\mathbb{E}_{\hat{s}_{t-h-d+1:t-h}}\left[\pi_{(c+d,h)}\Big|C\right], \pi^*)|C\right]$$
$$= \min_{\pi_{(c+d,h)}} \mathbb{E}_G\Big[\mathcal{L}(\hat{P}_{\text{train}}(g_{t-h-d+1:}^{t-h}|s_{t-h-d-c:}^{t-h-d}, a_{t-h-d:}^{t-h-1})\pi_{(c+d,h)}(a_t|\cdots, g_{t-h-d+1:}^{t-h}) +$$
$$\sum_{\substack{\hat{s}_{t-h-d+1:t-h} \\ \text{not all } g_{t'}}} \hat{P}_{\text{train}}(\hat{s}_{t-h-d+1:}^{t-h}|s_{t-h-d}, a_{t-h-d:}^{t-h-1}) \; \pi_{(c+d,h)}(a_t|\cdots, \hat{s}_{t-h-d+1:}^{t-h}), \pi^*)|C\Big]$$
$$\leq \min_{\pi_{(c+d,h)}} \mathbb{E}_G\Big[\mathcal{L}(\prod_{t' \in \tau_f} \hat{P}_f(t')\pi_{(c+d,h)}(a_t|\cdots, g_{t-h-d+1:}^{t-h}) +$$
$$\sum_{\substack{\hat{s}_{t-h-d+1:t-h} \\ \text{not all } g_{t'}}} \hat{P}_{\text{train}}(\hat{s}_{t-h-d+1:}^{t-h}|s_{t-h-d}, a_{t-h-d:}^{t-h-1}) \; \pi_{(c+d,h)}(a_t|\cdots, \hat{s}_{t-h-d+1:}^{t-h}), \pi^*)|C\Big]$$

where we computed the expectation $\mathbb{E}_{\hat{s}_{t-h-d+1:t-h}}\left[\pi_{(c+d,h)}\Big|C\right]$ by grouping into two terms : one where every $\hat{s}_{t-h-d+1:t-h} = g_{t-h-d+1:t-h}$ and one where there is at least one term $\hat{s}_i$ that is not $g_i$. This grouping was done using the definition of noise in our environment. We introduce the following notation here

$$\hat{P}_{\neq g_{t'}} := \sum_{\substack{\hat{s}_{t-h-d+1:t-h} \\ \text{not all } g_{t'}}} \hat{P}_{\text{train}}(\hat{s}_{t-h-d+1:}^{t-h}|s_{t-h-d}, a_{t-h-d:}^{t-h-1}).$$

Similarly,

$$P_{\neq g_{t'}} := \sum_{\substack{s_{t-h-d+1:t-h} \\ \text{not all } g_{t'}}} P_{\text{test}}(s_{t-h-d+1:}^{t-h}|s_{t-h-d}, a_{t-h-d:}^{t-h-1}).$$

With this notation, we continue our expansion:

$$\min_{\pi_{(c,h+d)}} \mathbb{E}_G \left[ \mathcal{L}(\pi_{(c,h+d)}, \pi^*) | C \right]$$

$$\leq \min_{\pi_{(c+d,h)}} \mathbb{E}_G \left[ \mathcal{L}( \prod_{t' \in \tau_f} \hat{P}_f(t') \pi_{(c+d,h)}(a_t | \cdots, g_{t-h-d+1:}^{t-h}) + \right.$$

$$\left. \sum_{\substack{\hat{s}_{t-h-d+1:t-h} \\ \text{not all } g_{t'}}} \hat{P}_{\text{train}}(\hat{s}_{t-h-d+1:}^{t-h} | s_{t-h-d}, a_{t-h-d:}^{t-h-1}) \; \pi_{(c+d,h)}(a_t | \cdots, \hat{s}_{t-h-d+1:}^{t-h}), \pi^*) | C \right]$$

$$\leq \min_{\pi_{(c+d,h)}} \mathbb{E}_G \left[ \mathcal{L}( \prod_{t' \in \tau_f} \hat{P}_f(t') \pi_{(c+d,h)}(a_t | \cdots, g_{t-h-d+1:}^{t-h}) + \hat{P}_{\neq g_{t'}} \; \pi_{(c+d,h)}(a_t | \cdots, \hat{s}_{t-h-d+1:}^{t-h}), \pi^*) | C \right]$$

$$\leq \min_{\pi_{(c+d,h)}} \mathbb{E}_G \left[ \prod_{t' \in \tau_f} \hat{P}_f(t') \mathcal{L}(\pi_{(c+d,h)}(a_t | \cdots, g_{t-h-d+1:}^{t-h}), \pi^*) | C \right]$$

$$+ \mathbb{E}_G \left[ \hat{P}_{\neq g_{t'}} \mathcal{L}(\pi_{(c+d,h)}(a_t | \cdots, \hat{s}_{t-h-d+1:}^{t-h}), \pi^*) | C \right]$$

where we got the inequality using the fact that $\mathcal{L}$ is a convex function and, thus, convex in each argument. Next, we take the expectation over $s_{t-h-d+1:t-h}$ by grouping the terms into two: one where every $s_{t-h-d+1:t-h} = g_{t-h-d+1:t-h}$ and one where there is at least one term $s_i \neq g_i$. Then, with some suppression of notation in the expression of $\pi_{(c+d,h)}$ and $G' \coloneqq G \setminus \{a_t, s_{t-h-d+1:t-h}\}$:

$$\min_{\pi_{(c,h+d)}} \mathbb{E}_G \left[ \mathcal{L}(\pi_{(c,h+d)}, \pi^*) | C \right]$$

$$\leq \min_{\pi_{(c+d,h)}} \mathbb{E}_{a_t}$$

$$\left[ \prod_{t' \in \tau_f} P_f^{\text{test}}(t') \, \hat{P}_f(t') \mathbb{E}_{G'} \left[ \mathcal{L}(\pi_{(c+d,h)}(\ldots \hat{s}_{t-h-d+1:}^{t-h} = g_{t-h-d+1:}^{t-h}), \pi^*) \Big| \ldots, s_{t-h-d+1:}^{t-h} = g_{t-h-d+1:}^{t-h} \right] \right.$$

$$+ P_{\neq g_{t'}} \prod_{t' \in \tau_f} \hat{P}_f(t') \mathbb{E}_{G'} \left[ \mathcal{L}(\pi_{(c+d,h)}(\ldots \hat{s}_{t-h-d+1:}^{t-h} = g_{t-h-d+1:}^{t-h}), \pi^*) \Big| \ldots, s_{t-h-d+1:}^{t-h} \neq g_{t-h-d+1:}^{t-h} \right]$$

$$+ \prod_{t' \in \tau_f} P_f^{\text{test}}(t') \hat{P}_{\neq g_{t'}} \mathbb{E}_{G'} \left[ \mathcal{L}(\pi_{(c+d,h)}(\ldots \hat{s}_{t-h-d+1:}^{t-h} \neq g_{t-h-d+1:}^{t-h}), \pi^*) \Big| \ldots, s_{t-h-d+1:}^{t-h} = g_{t-h-d+1:}^{t-h} \right]$$

$$+ P_{\neq g_{t'}} \hat{P}_{\neq g_{t'}} \mathbb{E}_{G'} \left[ \mathcal{L}(\pi_{(c+d,h)}(\ldots, \hat{s}_{t-h-d+1:}^{t-h} \neq g_{t-h-d+1:}^{t-h}), \pi^*) \Big| \ldots, s_{t-h-d+1:}^{t-h} \neq g_{t-h-d+1:}^{t-h} \right]$$

Now, we group all the terms into two - one representing where the learner's simulation matches the reality and one where it does not. Continuing from where we left off, first, define $\hat{P}_{\hat{s}=s}^f$ to be the probability that the inferred states at timesteps in $\tau_f$ are the states that were visited in reality. Then,

$$\min_{\pi_{(c,h+d)}} \mathbb{E}_G \left[ \mathcal{L}(\pi_{(c,h+d)}, \pi^*) | C \right]$$

$$\leq \min_{\pi_{(c+d,h)}} \mathbb{E}_{a_t}$$

$$\left[ \prod_{t' \in \tau_f} P_f^{\text{test}}(t') \, \hat{P}_f(t') \mathbb{E} \left[ \mathcal{L}(\pi_{(c+d,h)}(a_t | \ldots, \hat{s}_{t-h-d+1:}^{t-h} = g_{t-h-d+1:}^{t-h} = s_{t-h-d+1:}^{t-h}), \pi^*) \Big| \ldots, s_{t-h-d+1:}^{t-h} = g_{t-h-d+1:}^{t-h} \right] \right.$$

$$+ P_{\neq g_{t'}} \, \hat{P}_{\hat{s}=s}^f \mathbb{E} \left[ \mathcal{L}(\pi_{(c+d,h)}(a_t | \ldots, \hat{s}_{t-h-d+1:}^{t-h} = s_{t-h-d+1:}^{t-h} \neq g_{t-h-d+1:}^{t-h}), \pi^*) \Big| \ldots, s_{t-h-d+1:}^{t-h} \neq g_{t-h-d+1:}^{t-h} \right]$$

$$+ \prod_{t' \in \tau_f} P_f^{\text{test}}(t') \hat{P}_{\neq g_{t'}} \mathbb{E} \left[ \mathcal{L}(\pi_{(c+d,h)}(a_t | \ldots, \hat{s}_{t-h-d+1:}^{t-h} \neq g_{t-h-d+1:}^{t-h} = s_{t-h-d+1:}^{t-h}), \pi^*) \Big| \ldots, s_{t-h-d+1:}^{t-h} = g_{t-h-d+1:}^{t-h} \right]$$

$$+ P_{\neq g_{t'}} \prod_{t' \in \tau_f} \hat{P}_f(t') \mathbb{E} \left[ \mathcal{L}(\pi_{(c+d,h)}(a_t | \ldots, \hat{s}_{t-h-d+1:}^{t-h} = g_{t-h-d+1:}^{t-h} \neq s_{t-h-d+1:}^{t-h}), \pi^*) \Big| \ldots, s_{t-h-d+1:}^{t-h} \neq g_{t-h-d+1:}^{t-h} \right]$$

$$+ P_{\neq g_{t'}} \hat{P}_{\text{train}}(\hat{s}_{t-h-d+1:}^{t-h} \neq s_{t-h-d+1:}^{t-h}; g_{t-h-d+1:}^{t-h} | s_{t-h-d}, a_{t-h-d:}^{t-h-1})$$

$$\mathbb{E} \left[ \mathcal{L}(\pi_{(c+d,h)}(a_t | \ldots, \hat{s}_{t-h-d+1:}^{t-h} \neq s_{t-h-d+1:}^{t-h} \neq g_{t-h-d+1:}^{t-h}), \pi^*) \Big| \ldots, s_{t-h-d+1:}^{t-h} \neq g_{t-h-d+1:}^{t-h} \right] | C, a_t \left. \right] | |C]$$

For the match terms, we use the fact that $\prod_{t'\in\tau_f} P_f(t') \leq 1$ and $\hat{P}^f_{\hat{s}=s} = \hat{P}_{\text{train}}(\hat{s}^{t-h}_{t-h-d+1:} = s^{t-h}_{t-h-d+1:}|s_{t-h-d}, a^{t-h-1}_{t-h-d:}) \leq 1$. For the mismatch terms, we use the definition of $\epsilon_f$ and Lemma 5. Then, we continue:

$$\leq \min_{\pi_{(c+d,h)}} \mathbb{E}_G \left[ \mathcal{L}(\pi_{(c+d,h)}, \pi^*) \Big| C \right] \qquad \text{(Simulation matches reality)}$$
$$+ \prod_{t'\in\tau_f} P^{\text{test}}_f(t') \left[ (1 - \hat{P}_f(t'))\epsilon_f \right] + \prod_{t'\in\tau_f} (1 - P^{\text{test}}_f(t')) \left[ \hat{P}_f(t') + \hat{P}_{\neq g_{t'}, s_{t'}} \right] \epsilon_f.$$
$$\text{(Simulation does not match reality)}$$

Recall that we write $P^{\text{test}}_f(t')$ as simply $P_f(t')$. We simplify the mismatch terms further:

$$\prod_{t'\in\tau_f} P_f(t') \left[ (1 - \hat{P}_f(t'))\epsilon_f \right] + \prod_{t'\in\tau_f} (1 - P_f(t')) \left[ \hat{P}_f(t') + \hat{P}_{\neq g_{t'}, s_{t'}} \right] \epsilon_f$$
$$\leq \prod_{t'\in\tau_f} P_f(t') \left[ (1 - \hat{P}_f(t'))\epsilon_f \right] + \prod_{t'\in\tau_f} (1 - P_f(t')) \left[ \hat{P}_f(t') + (1 - \hat{P}_f(t')) \right] \epsilon_f$$
$$= \epsilon_f \cdot \left[ 1 - \prod_{t'\in\tau_f} P_f(t')\hat{P}_f(t') \right].$$
$$= \epsilon_f \cdot \left[ 1 - \prod_{t'\in\tau_f} P_f(t')(P_f(t') + \delta_f(t')) \right].$$

Next, we simplify the match terms by using the definition of $\alpha_b$:

$$\min_{\pi_{(c+d,h)}} \mathbb{E}_G \left[ \mathcal{L}(\pi_{(c+d,h)}, \pi^*) \Big| C \right] = \min_{\pi_{(c,h)}} \mathbb{E}_G \left[ \mathcal{L}(\pi_{(c,h)}, \pi^*) | C \right] - \alpha_b.$$

Substituting these two terms back in, we conclude.

Now, we prove the lower bound. We proceed in a manner similar to the proof of the upper bound. For ease of notation, we will write $x^b_{a:}$ to mean $x_{a:b}$. Additionally, for greater clarity, we will explicitly include the context length of each model, so $\pi_{(c,h)} = \pi_h$ and $\pi_{(c,h+d)} = \pi_{h+d}$. We start by writing, using Assumption 1,

$$\min_{\pi_{(c,h)}} \mathbb{E}_G \left[ \mathcal{L}(\pi_{(c,h)}, \pi^*) \mid C \right]$$

$$= \min_{\pi_{(c+d,h)}} \mathbb{E}_G \left[ \mathcal{L}(\hat{P}_{\text{train}}(g^{t-h-c-1}_{t-h-d-c:}|s_{t-h-c})\pi_{(c+d,h)} + \sum_{\substack{\hat{s}^{t-h-c-1}_{t-h-d-c:}, \\ \text{not all } g_{t'}}} \hat{P}_{\text{train}}(\hat{s}^{t-h-c-1}_{t-h-d-c:}|s_{t-h-c})\pi_{(c+d,h)}, \pi^*) \Big| C \right].$$

$$\leq \min_{\pi_{(c+d,h)}} \mathbb{E}_G \left[ \prod_{t'\in\tau_b} \hat{P}_b(t')\mathcal{L}(\pi_{(c+d,h)}, \pi^*) + \sum_{\substack{\hat{s}^{t-h-c-1}_{t-h-d-c:}, \\ \text{not all } g_{t'}}} \hat{P}_{\text{train}}(\hat{s}^{t-h-c-1}_{t-h-d-c:}|s_{t-h-c})\mathcal{L}(\pi_{(c+d,h)}, \pi^*) \Big| C \right].$$

We introduce the following notation here

$$\hat{P}_{\neq g_{t'}} := \sum_{\substack{\hat{s}_{t-h-d-c:t-h-c-1} \\ \text{not all } g_{t'}}} \hat{P}_{\text{train}}(\hat{s}^{t-h-c-1}_{t-h-d-c:}|s_{t-h-c}).$$

Similarly,

$$P_{\neq g_{t'}} := \sum_{\substack{s_{t-h-d-c:t-h-c-1} \\ \text{not all } g_{t'}}} P_{\text{test}}(s^{t-h-c-1}_{t-h-d-c:}|s_{t-h-c}).$$

With this notation, we continue our expansion:

$$\min_{\pi_{(c,h)}} \mathbb{E}_G \left[ \mathcal{L}(\pi_{(c,h)}, \pi^*)|C \right]$$

$$\leq \min_{\pi_{(c+d,h)}} \mathbb{E}_G \left[ \prod_{t' \in \tau_b} \hat{P}_b(t') \mathcal{L}(\pi_{(c+d,h)}(a_t|..., g_{t-h-d-c:}^{t-h-c-1}), \pi^*) \mid C \right] +$$

$$\mathbb{E}_G \left[ \hat{P}_{\neq g_{t'}} \mathcal{L}(\ \pi_{(c+d,h)}(a_t|..., \hat{s}_{t-h-d-c:}^{t-h-c-1} \neq g_{t-h-d-c:}^{t-h-c-1}), \pi^*) \mid C \right]$$

where we got the inequality using the fact that $\mathcal{L}$ is a convex function. Next, we take the expectation over $s_{t-h-d-c:t-h-c-1}$ by grouping the terms into two: one where every $s_{t-h-d-c:t-h-c-1} = g_{t-h-d-c:t-h-c-1}$ and one where there is at least one term $s_i \neq g_i$. Then, again suppressing some terms inside the expression of $\pi_{(c+d,h)}$:

$$\min_{\pi_{(c,h)}} \mathbb{E}_G \left[ \mathcal{L}(\pi_{(c,h)}, \pi^*)|C \right]$$

$$\leq \min_{\pi_{(c+d,h)}} \mathbb{E}_{a_t}$$

$$\left[ \prod_{t' \in \tau_b} P_b(t')\hat{P}_b(t') \ \mathbb{E} \left[ \mathcal{L}(\pi_{(c+d,h)}(..., \hat{s}_{t-h-d-c:}^{t-h-c-1} = g_{t-h-d-c:}^{t-h-c-1} = s_{t-h-d-c:}^{t-h-c-1}), \pi^*) \Big|..., s_{t-h-d-c:}^{t-h-c-1} = g_{t-h-d-c:}^{t-h-c-1} \right] \right.$$

$$+ P_{\neq g_{t'}} \prod_{t' \in \tau_b} \hat{P}_b(t') \mathbb{E} \left[ \mathcal{L}(\pi_{(c+d,h)}(..., \hat{s}_{t-h-d-c:}^{t-h-c-1} = g_{t-h-d-c:}^{t-h-c-1} \neq s_{t-h-d-c:}^{t-h-c-1}), \pi^*) \Big|..., s_{t-h-d-c:}^{t-h-c-1} \neq g_{t-h-d-c:}^{t-h-c-1} \right]$$

$$+ \prod_{t' \in \tau_b} P_b(t')\hat{P}_{\neq g_{t'}} \ \mathbb{E} \left[ \mathcal{L}(\pi_{(c+d,h)}(..., \hat{s}_{t-h-d-c:}^{t-h-c-1} \neq g_{t-h-d-c:}^{t-h-c-1} = s_{t-h-d-c:}^{t-h-c-1}), \pi^*) \Big|..., s_{t-h-d-c:}^{t-h-c-1} = g_{t-h-d-c:}^{t-h-c-1} \right]$$

$$\left. + P_{\neq g_{t'}} \hat{P}_{\neq g_{t'}} \ \mathbb{E} \left[ \mathcal{L}(\pi_{(c+d,h)}(..., \hat{s}_{t-h-d-c:}^{t-h-c-1} \neq g_{t-h-d-c:}^{t-h-c-1}), \pi^*) \Big|..., s_{t-h-d-c:}^{t-h-c-1} \neq g_{t-h-d-c:}^{t-h-c-1} \right] \mid C, a_t \right] \mid C \right]$$

Now, we group all the terms into two - one representing where the learner's simulation matches the reality and one where it does not. Continuing from where we left off and defining $\hat{P}_{\hat{s}=s}^b := \hat{P}_{\text{train}}(\hat{s}_{t-h-d-c:}^{t-h-c-1} = s_{t-h-d-c:}^{t-h-c-1}|s_{t-h-c})$:

$$\min_{\pi_{(c,h)}} \mathbb{E}_G \left[ \mathcal{L}(\pi_{(c,h)}, \pi^*)|C \right]$$

$$\leq \min_{\pi_{(c+d,h)}} \mathbb{E}_{a_t}$$

$$\left[ \prod_{t' \in \tau_b} P_b(t') \ \hat{P}_b(t') \mathbb{E} \left[ \mathcal{L}(\pi_{(c+d,h)}(a_t|..., \hat{s}_{t-h-d-c:}^{t-h-c-1} = g_{t-h-d-c:}^{t-h-c-1}), \pi^*) \Big|..., s_{t-h-d-c:}^{t-h-c-1} = g_{t-h-d-c:}^{t-h-c-1} \right] \right.$$

$$+ P_{\neq g_{t'}} \ \hat{P}_{\hat{s}=s}^b \mathbb{E} \left[ \mathcal{L}(\pi_{(c+d,h)}(a_t|..., \hat{s}_{t-h-d-c:}^{t-h-c-1} = s_{t-h-d-c:}^{t-h-c-1}), \pi^*) \Big|..., s_{t-h-d-c:}^{t-h-c-1} \neq g_{t-h-d-c:}^{t-h-c-1} \right]$$

$$+ \prod_{t' \in \tau_b} P_b(t')\hat{P}_{\neq g_{t'}} \ \mathbb{E} \left[ \mathcal{L}(\pi_{(c+d,h)}(a_t|..., s_{t-h-d-c:}^{t-h-c-1} \neq g_{t-h-d-c:}^{t-h-c-1}), \pi^*) \Big|..., s_{t-h-d-c:}^{t-h-c-1} = g_{t-h-d-c:}^{t-h-c-1} \right]$$

$$+ P_{\neq g_{t'}} \prod_{t' \in \tau_b} \hat{P}_b(t') \ \mathbb{E} \left[ \mathcal{L}(\pi_{(c+d,h)}(a_t|..., \hat{s}_{t-h-d-c:}^{t-h-c-1} = g_{t-h-d-c:}^{t-h-c-1}), \pi^*) \Big|..., s_{t-h-d-c:}^{t-h-c-1} \neq g_{t-h-d-c:}^{t-h-c-1} \right] \mid C, a_t \right] \mid C \right]$$

$$+ P_{\neq g_t} \hat{P}_{\text{train}}(\hat{s}_{t-h-d-c:}^{t-h-c-1} \neq g_{t-h-d-c:}^{t-h-c-1}, s_{t-h-d-c:}^{t-h-c-1} \mid s_{t-h-c})$$

$$\mathbb{E} \left[ \mathcal{L}(\pi_{(c+d,h)}(a_t|..., \hat{s}_{t-h-d-c:}^{t-h-c-1} \neq s_{t-h-d-c:}^{t-h-c-1}), \pi^*) \Big|..., s_{t-h-d-c:}^{t-h-c-1} \neq g_{t-h-d-c:}^{t-h-c-1} \right] \mid C, a_t \right] \mid C \right]$$

For the match terms, we use the fact that $\prod_{t' \in \tau_b} \hat{P}_b(t') \leq 1$ and $\hat{P}_{\text{train}}(\hat{s}_{t-h-d-c:}^{t-h-c-1} = s_{t-h-d-c:}^{t-h-c-1}|s_{t-h-c:}^{t-h}) \leq 1$. For the mismatch terms, we use the definition of $\epsilon_b$

and Lemma 5. Then, we continue:

$$\leq \min_{\pi_{(c+d,h)}} \mathbb{E}_G \left[ \mathcal{L}(\pi_{(c+d,h)}, \pi^*) \Big| C \right] \qquad \text{(Simulation matches reality)}$$

$$+ \prod_{t' \in \tau_b} P_b(t')(1 - \hat{P}_b(t'))\epsilon_b + \prod_{t' \in \tau_b} (1 - P_b(t'))(\hat{P}_b(t') + \hat{P}_{\neq g_{t'}, s_{t'}})\epsilon_b.$$

$$\text{(Simulation does not match reality)}$$

We simplify the mismatch terms further:

$$\prod_{t' \in \tau_b} P_b(t')(1 - \hat{P}_b(t'))\epsilon_b + \prod_{t' \in \tau_b} (1 - P_b(t'))(\hat{P}_b(t') + \hat{P}_{\neq g_{t'}, s_{t'}})\epsilon_b$$

$$\leq \prod_{t' \in \tau_b} P_b(t')(1 - \hat{P}_b(t'))\epsilon_b + \prod_{t' \in \tau_b} (1 - P_b(t'))(\hat{P}_b(t') + (1 - \hat{P}_b(t')))\epsilon_b$$

$$= \epsilon_b \cdot \left[ 1 - \prod_{t' \in \tau_b} P_b(t')\hat{P}_b(t') \right]$$

$$= \epsilon_b \cdot \left[ 1 - \prod_{t' \in \tau_b} P_b(t')(P_b(t') + \delta_b(t')) \right]$$

.

Next, we simplify the match terms by using the definition of $\alpha_f$ which follows from Proposition 7 in a manner similar to the definition of $\alpha_b$:

$$\min_{\pi_{(c+d,h)}} \mathbb{E}_G \left[ \mathcal{L}(\pi_{(c+d,h)}, \pi^*) \Big| C \right]$$

$$= \min_{\pi_{(c,h+d)}} \mathbb{E}_G \left[ \mathcal{L}(\pi_{(c,h+d)}, \pi^*) | C \right] - \alpha_f. \qquad \text{(Proposition 7)}$$

We substitute these terms back in to get the desired bound. $\qquad \square$

Now, we prove Corollary 2 as a direct consequence of the Consistency-Reactivity Inequalities. Recall that in a near-deterministic environment, $P_f$ is close to 1 as the transitions are purely determined.

**Corollary 2** (Consistency). Suppose, the train and test environments are the same and it is deterministic. Suppose $a_t$ is influenced by at least one state at time steps $\tau_b$ and let $\delta_f(t') \approx 0$ for all $t' \in \tau_f$. If the diversity in past strategies is not 0, and $\epsilon_f$ is finite, then

$$\min_{\pi_{h+d}} \mathbb{E}_G \left[ \mathcal{L}(\pi_{h+d}, \pi^*) | C \right] < \min_{\pi_h} \mathbb{E}_G \left[ \mathcal{L}(\pi_h, \pi^*) | C \right].$$

*Proof.* This follows from the upper bound of Proposition 1. Since the test environment is deterministic, we take $P_f(t') \approx 1$ and, by hypothesis, $\delta_f(t') \approx 0$ for all $t' \in \tau_f$, menaing our learned policy's implicit dynamics model is accurate. Then, we get

$$\min_{\pi_{h+d}} \mathbb{E}_G \left[ \mathcal{L}(\pi_{h+d}, \pi^*) \mid C \right] - \min_{\pi_h} \mathbb{E}_G \left[ \mathcal{L}(\pi_h, \pi^*) \mid C \right] \leq -\alpha_b + \epsilon_f (1 - \prod_{t' \in \tau_f} P_f(t')(P_f(t') + \delta_f(t'))) = -\alpha_b$$

since $\epsilon_f$ is finite. All that remains to show is that $\alpha_b$ is positive. Note that $a_t$ is temporally dependent on at least one state in $\{s_{t-h-c-d:t-h-c-1}\}$. Furthermore, since diversity in past strategies is not 0, the states in timesteps $\tau_b$ cannot be predicted with probability 1. Then, by Proposition 6, we have that $\alpha_b > 0$. Therefore,

$$\min_{\pi_{h+d}} \mathbb{E}_G \left[ \mathcal{L}(\pi_{h+d}, \pi^*) \mid C \right] - \min_{\pi_h} \mathbb{E}_G \left[ \mathcal{L}(\pi_h, \pi^*) \mid C \right] \leq -\alpha_b < 0.$$

$\qquad \square$

Next, we prove Corollary 3.

**Corollary 3** Suppose $P_f(t')$ is small or both $P_f(t')$ and $|\delta_f(t')|$ are large for all $t' \in \tau_f$. If temporal dependency decreases over time such that $\epsilon_b$ is small and $a_t$ is influenced by at least one state at time steps in $\tau_f$, then

$$\min_{\pi_{h+d}} \mathbb{E}_G\left[\mathcal{L}(\pi_{h+d}, \pi^*)|C\right] > \min_{\pi_h} \mathbb{E}_G\left[\mathcal{L}(\pi_h, \pi^*)|C\right].$$

*Proof.* Notice that since $\epsilon_b \approx 0$, the lower bound in Proposition 1 is $\alpha_f$. Next, we show that that $\alpha_f$ is positive. If $P_f(t')$ is small, then we know by Proposition 7 that $\alpha_f$ is positive. On the other hand, if $P_f(t') \approx 1$ but $|\delta_f(t')|$ is large, then since probabilities are bounded, we require $\delta_f(t') \approx -1$. In other words, the inference of the correct unobserved state is very unlikely causing $\alpha_f$ to be positive again. Therefore,

$$\min_{\pi_h} \mathbb{E}_G\left[\mathcal{L}(\pi_h - \pi^*)|C\right] < \min_{\pi_{h+d}} \mathbb{E}_G\left[\mathcal{L}(\pi_{h+d} - \pi^*)|C\right]$$

$\square$

To sanity check the Consistency-Reactivity Inequalities, we prove the next proposition. This shows that the right-hand side of the inequalities is greater than or equal to the left-hand side.

**Proposition 9.** *In all environments,*

$$-\epsilon_b(1 - \prod_{t' \in \tau_b} P_b(t')(P_b(t') + \delta_b(t'))) \leq -\alpha_b$$

*and*

$$\alpha_f \leq \epsilon_f(1 - \prod_{t' \in \tau_f} P_f(t')(P_f(t') + \delta_f(t'))).$$

*Proof.* We prove the first inequality; the proof of the second proceeds similarly. Using the definition of $\alpha_b$ and suppressing $G$ and $C$ (as in Proposition 1) for clarity, we have:

$$\begin{aligned}
\alpha_b = & -\min_{\pi_{h+d}} \mathbb{E}[\mathcal{L}(\pi_{h+d}, \pi^*)] + \min_{\pi_h} \mathbb{E}[\mathcal{L}(\pi_h, \pi^*)] \\
\leq & -\min_{\pi_{h+d}} \mathbb{E}[\mathcal{L}(\pi_{h+d}, \pi^*)] \\
& + \min_{\pi_{h+d}} \mathbb{E}\Big[\mathcal{L}\Big(P_{\text{correct inference}}\pi_{h+d}(\text{correct inference}) \\
& + P_{\text{incorrect inference}}\pi_{h+d}(\text{incorrect inference}), \pi^*\Big)\Big] \\
\leq & -\min_{\pi_{h+d}} \mathbb{E}[\mathcal{L}(\pi_{h+d}, \pi^*)] \\
& + \min_{\pi_{h+d}} \mathbb{E}\Big[P_{\text{correct inference}}\mathcal{L}\big(\pi_{h+d}(\text{correct inference}), \pi^*\big) \\
& + P_{\text{incorrect inference}}\mathcal{L}\big(\pi_{h+d}(\text{incorrect inference}), \pi^*\big)\Big] \\
\leq & -\min_{\pi_{h+d}} \mathbb{E}[\mathcal{L}(\pi_{h+d}, \pi^*)] \\
& + \min_{\pi_{h+d}}\Big(\mathbb{E}\big[\mathcal{L}\big(\pi_{h+d}(\text{correct inference}), \pi^*\big)\big]\Big) \\
& + \mathbb{E}\big[P_{\text{incorrect inference}}\mathcal{L}\big(\pi_{h+d}(\text{incorrect inference}), \pi^*\big)\big] \\
\leq & -\min_{\pi_{h+d}} \mathbb{E}[\mathcal{L}(\pi_{h+d}, \pi^*)] \\
& + \min_{\pi_{h+d}}\Big(\mathbb{E}\big[\mathcal{L}\big(\pi_{h+d}(\text{correct inference}), \pi^*\big)\big]\Big) \\
& + \epsilon_b(1 - \prod_{t' \in \tau_b} P_b(t')(P_b(t')\delta_b(t')))
\end{aligned}$$

$$= \epsilon_b(1 - \prod_{t' \in \tau_b} P_b(t')(P_b(t') + \delta_b(t')))$$

$\square$

### E.4 CLOSED-LOOP VERSUS OPEN-LOOP IN HIGHLY STOCHASTIC AND NEAR-DETERMINISTIC ENVIRONMENTS

The Consistency-Reactivity Inequalities allow us to make an even stronger statement when we compare strictly closed-loop policies with open-loop ones. Consider the same set-up as before with $h = 0$. Thus, $\pi_{(c,0)}$ represents a closed-loop policy whereas $\pi_{(c,d)}$ represents an open-loop one. We can compare these policies' divergences with the expert across the entire trajectory in the limiting cases of the environment stochasticity.

**Corollary 10.** *Suppose $P_f(t')$ is small or both $P_f(t')$ and $|\delta_f(t')|$ are large for all $t' \in \tau_f$. If temporal dependency decreases over time such that $\epsilon_b$ is small and $a_t$ is influenced by at least one state at time steps in $\tau_f$, then the expected divergence between the closed-loop policy over the full trajectory is lower than that between the open-loop policy and the expert.*

*Suppose, the train and test environments are the same and it is deterministic. Suppose $a_t$ is influenced by at least one state at time steps $\tau_b$ and let $\delta_f(t') \approx 0$ for all $t' \in \tau_f$. If the diversity in past strategies is not 0, and $\epsilon_f$ is finite, then the divergence between the closed-loop policy over the full trajectory is greater than that between the open-loop policy and the expert.*

*Proof.* At any arbitrary time step $t$, the chunks of the two policies can be aligned in one of two ways:

Case 1: $\pi_{(c,0)}$ is executing $a_t$ as the first action in its action chunk and $\pi_{(c,d)}$ is also executing $a_t$ as the first action in its action chunk.

Case 2: $\pi_{(c,0)}$ is executing $a_t$ as the first action in its action chunk and $\pi_{(c,d)}$ is executing $a_t$ as the $k$-th action, where $k \in (1, 1 + d]$ in its action chunk.

Using the Consistency-Reactivity Inequalities, in Case 1, both policies have equal divergence. However, in case 2, using Corollary 3, we know that the closed-loop policy will outperform the open-loop one in the first setting of the statement and open-loop will outperform in the second. From this we can conclude the divergence across the full trajectory. $\square$

