# OpenReview forum: "Bidirectional Decoding: Improving Action Chunking via Guided Test-Time Sampling"
_ICLR.cc/2025/Conference — ICLR 2025 Poster_

### Official Review · Reviewer_wSU2 · 2024-10-27

**Soundness:** 3
**Presentation:** 1
**Contribution:** 2
**Rating:** 8
**Confidence:** 3

**Summary:**

The study presents a theoretical exploration of action chunking. It offers valuable insights, demonstrating the feature of action chunking varies across different scenarios, particularly depending on the degree of determinism or stochasticity in the environment. The findings suggest that in deterministic environments, action chunking improves long-term consistency, whereas in highly uncertain environments, it struggles to adapt effectively to changes, as reactivity becomes more critical.

The paper introduces BID—an inference-time, model-agnostic, computationally efficient, and easy-to-use plug-and-play algorithm designed to enhance action chunking, emphasizing better closed-loop control. BID selects action chunks from samples based on two key criteria: backward coherence and forward contrast.

The experiments begin with a simple yet effective discrete one-dimensional toy policy, validating the theoretical insights on the relationship between action chunking and varying degrees of environmental determinism or stochasticity. The effectiveness of BID is further evaluated using Diffusion Policy as the base policy and compared against three other baseline action chunking strategies (vanilla, warm start, and EMA) across three simulators: Push T (single task), RoboMimic (four tasks: Lift, Can, Square, and Transport), and Kitchen.

Additional experiments explore BID’s scalability by varying batch sizes and its compatibility with different sampling strategies. The results also demonstrate BID’s generality and efficacy in real-world applications, reinforcing its practical utility.

**Strengths:**

The work employs well-founded models and thoughtful improvements to make a commendable attempt at uncovering the essential latent factors of action chunking, such as the environment’s degree of determinism. It offers insightful explanations of the theoretical analysis, clearly highlighting the benefits (consistency) and drawbacks (reduced reactivity) of action chunking.

The experiments are comprehensive, demonstrating BID’s effectiveness across a variety of scenarios.

**Weaknesses:**

I found some of the figures difficult to interpret. For example, in Figure 1, I assume the goal is to illustrate how BID combines the benefits of (a) and (b). However, the concepts introduced in the paper, such as backward coherence and forward contrast, are not intuitively reflected in the figure. Similarly, the meaning of Figure 3 is unclear.

A more significant concern is the lack of a clear connection between the BID criteria in Section 4.2 and the intended trade-off between consistency and reactivity. Specifically, the discussion on reactivity is vague, and it is not evident how the BID design enhances reactivity. I believe additional explanations are necessary to clarify how the proposed BID method addresses reactivity, providing more detailed reasoning behind the design choices.

**Questions:**

I have a detailed question regarding the design of backward coherence
$\mathcal{L}_B$ and forward contrast $\mathcal{L}_F$. Specifically, you use Euclidean distance to evaluate the consistency (similarity) of actions at individual time steps. However, there are many alternative metrics that might be more suitable depending on the nature of the values. For example, in typical 7-DoF robotic arm actions, three values represent orientation, and Euclidean distance may not be the most appropriate metric. In some cases, L1 distance or angle-based metrics (e.g., cosine similarity) might be more suitable or even more direct alternatives. Could you provide additional experiments to explore the impact of different metrics on your two criteria and their effect on the experimental outcomes?

I am also curious about the design of forward contrast. I did not clearly see its connection to your theoretical analysis, particularly in terms of its relationship with reactivity. Could you provide more insights into why you designed forward contrast this way and how it specifically enhances reactivity? Alternatively, if there are other possible designs that improve reactivity, I suggest revisiting and reworking Sec.4.2 to clarify these aspects more thoroughly.

Additionally, I am wondering whether the weighted sampling strategy used in ACT (introduced with [ALOHA](https://arxiv.org/abs/2304.13705)) could serve as a reasonable baseline for comparison. It would be helpful to see comparison results between BID and action sampling strategy used in ACT. Could you conduct further experiments to evaluate this comparison and enhance the analysis of BID’s effectiveness?

---

> ### Author Response · Authors · 2024-11-22
>
> Thank you for your thoughtful feedback. Please find our response to your comments below.
>
> > It would be helpful to see comparison results between BID and action sampling strategy used in ACT.
> * We have compared them in Sec 5.2. Specifically, the EMA baseline is the temporal ensembling technique used in ACT. Our experiments show that BID generally outperforms EMA in both simulation tasks (Fig 6, Tab 4) and real-world tasks (Fig 9). Moreover, BID can be combined with EMA for complementary benefits (Fig 7).
>
> > There are alternative metrics that might be more suitable. Could you provide additional experiments to explore the impact of different metrics?
> * Thank you for your suggestion. We have added experiments to evaluate our method using other distance metrics (L1 and cosine distance). The table below summarizes the results on three representative tasks. We observe that BID yields significant performance gains across all metrics, with L1 distance sometimes outperforming L2. We have included these new experimental results in Appendix A6.
>
>   |               | Square              | Lift               | Kitchen            |
>   |---------------|---------------------|--------------------|--------------------|
>   | Vanilla   | 0.68 $\pm$ 0.06    | 0.12 $\pm$ 0.02    | 0.22 $\pm$ 0.04    |
>   | BID L2        | **0.76 $\pm$ 0.03** | 0.58 $\pm$ 0.06    | 0.64 $\pm$ 0.05    |
>   | BID L1        | **0.76 $\pm$ 0.04** | **0.68 $\pm$ 0.05**| **0.70 $\pm$ 0.04**|
>   | BID Cosine    | 0.73 $\pm$ 0.04    | **0.70 $\pm$ 0.02**| 0.61 $\pm$ 0.03    |
>
>
> > Lack of a clear connection between the BID criteria and the trade-off. It is not evident how the BID design enhances reactivity.
> * Our analysis shows that shorter action horizons improve reactivity but compromise consistency. This analysis inspires our design of BID in two ways: (i) BID operates with a short horizon to maximize reactivity, and (ii) BID leverages the bidirectional criteria to select a sample that preserves consistency.
>
> > Could you provide more insights into why you designed forward contrast this way?
> * Forward contrast is designed to identify the *optimal sample* for the future plan. This is crucial when unexpected changes occur at the current time step.
> * To achieve this, we approximately maximize the likelihood of the selected sample under the expert policy by using a pair of learned policies. The strong policy proposes positive candidates for likelihood maximization, while the weak policy generates negative samples to address the discrepancy between the strong policy and expert policy.
> * Our initial manuscript illustrated the intuition behind this approach in Appendix B. To enhance clarity, we have expanded the discussion in Section 4.3 of the main draft and included additional ablation studies on VQ-BET in Appendix A5.
>
> > I found some of the figures difficult to interpret. For example, the concepts introduced in the paper are not intuitively reflected in Figure 1. Similarly, the meaning of Figure 3 is unclear.
> * Figure 1 was designed to illustrate the properties of different inference methods, rather than to directly describe the technical solution proposed in our method. We have clarified this purpose in the caption.
> * Fig 3 was designed to highlight the difference between the long-context policy, short-horizon policy and long-horizon policy. To improve visual clarity, we have revised the figure in our manuscript.

---

> > ### Comment · Reviewer_wSU2 · 2024-11-24
> >
> > Thank you for your response. I believe most of my concerns have been addressed.
> >
> > I have a suggestion regarding the design of BID. It seems that current research often focuses on different embodiments (e.g., humanoid, quadruped, etc.). My Q.1 was related to this issue, as in real-world scenarios, there are numerous robotic embodiments. Based on your methodology, I am unsure whether the 2-norm is universally optimal for all embodiments. This limitation might constrain generalization, as different metrics would need to be tested for various embodiments.
> >
> > Would it be possible to apply BID in the embedding space instead of directly on the action output to demonstrate its effectiveness? If BID proves effective in the embedding space, your method could potentially extend to foundation models, enabling seamless generalization across diverse robotic platforms. If time permits, I would be very interested in seeing experiments that evaluate BID’s performance in the latent embedding space rather than solely on action outputs.

---

> ### Author Response · Authors · 2024-11-25
>
> Thank you for your thoughtful feedback and suggestion.
>
> - To address your comment, we conducted an additional experiment with VQ-BeT, applying BID in its *latent embedding* space. The table below summarizes the success rate of closed-loop BID on the Push-T task, where BID with L2 distance in the latent embedding space slightly outperforms its counterpart in the output action space.
> - It’s worth noting that extending this design choice to a broader class of policies can be non-trivial. Many generative policies, including recent robotic foundation models [1,2], are based on diffusion or flow matching that lack explicit embedding spaces.
> - Overall, our default implementation of BID is model-agnostic, applicable across different policies, and demonstrates strong performance, while other model-specific distance metrics or embedding spaces might offer further improvement.
>
>   | Method               | Success Rate |
>   | -------------------- |:------------:|
>   | Vanilla              | 48.9 ± 2.7   |
>   | EMA                  | 52.6 ± 2.9   |
>   | BID Output Distance | 54.4 ± 1.8   |
>   | BID Latent Distance | 55.9 ± 1.5   |
>
> We hope these experiments and comparisons help clarify the strength and potential of our proposed method.
>
> [1] Octo: An Open-Source Generalist Robot Policy, 2024 \
> [2] π0: A Vision-Language-Action Flow Model for General Robot Control, 2024

---

> > ### Comment · Reviewer_wSU2 · 2024-11-26
> >
> > Thank you for your response. Overall, I find your work to be an interesting technique for improving inference time in robotic policy, an area that has been somewhat overlooked in robotics but is increasingly studied due to the prevalence of LLMs. While this work has been tested on a relatively narrow range of scenarios, I believe it deserves broader recognition so that others can test and further refine it in future research. As such, I will raise my score, driven by my own interest in the topic and its potential impact.

---

> ### Author Response · Authors · 2024-11-26
>
> We appreciate your encouraging comments, e.g., `interesting technique`, `overlooked in robotics`, and `deserves broader recognition`.
>
> For the community to `test and further refine it in future research`, we have released our code in both the LeRobot and Diffusion codebases, as detailed in the supplementary material.
> We hope this will facilitate further exploration beyond the *10 scenarios (1 synthetic, 7 simulated, 2 real-world)* examined in our work.
>
> Thank you again for your thoughtful engagement and review.

---

### Official Review · Reviewer_FR7d · 2024-11-02

**Soundness:** 3
**Presentation:** 4
**Contribution:** 3
**Rating:** 8
**Confidence:** 4

**Summary:**

This paper studies the usage of action chunking when learning policies through behavior cloning of human demonstration data on robots. Action chunking is the prediction and execution of a sequence of actions over a horizon $h$ without closed loop control. Long sequences of predicted actions lead to increased temporal consistency in the policy but lack the ability to react to short-term noise in a given system; on the other hand, short sequences of action chunks allow for this reactivity but at the expense of temporal stability. This is a well-known phenomena that affects the design decisions of machine learning and robotics practitioners. This paper provides theoretical analysis of the problem and introduce bidirectional decoding, an inference time policy change that naturally balances the need for long-term consistency with short term stability. The authors test their method on various toy and scaled up domains and show their method superior.

**Strengths:**

This paper is written clearly and well-motivated. To my knowledge, this is the first formalism of the optimality gaps incurred when trading off context length and prediction horizon. They demonstrate some exciting results that attempt to dissolve the issue of deciding an action-horizon on multiple domains like FrankaKitchen and PushT. They also describe some real-world experiments on robots but they are not the main focus of this paper.

**Weaknesses:**

There are several weak points in the paper that detract from the paper's quality.

1. **Human latents**: There is some motivation surrounding the latent intentions of humans that are not directly observed. However, talk of latent intentions is not mentioned in any technical sense so it is not clear how different latents $z_t$ affect the methodology and analysis presented in the paper
2. Lack of model-based/model-free discussion: In section 3.2, the authors outline their approach to theoretical analysis. It is not clear how model-free or model-based policies fit into this framework or the analysis. In my understanding of the analysis, the authors expect the policies to learn implicit dynamics models and do not ever explicitly learn them.
3. Ambiguous definitions in 3.2: I understand that the appendix provides better definitions of the quantities $\alpha$ and $\epsilon$, but given that they are crucial to understanding the analysis, the definitions in the main paper are not sufficient for understanding.
4. Difference between Corollary 2 and Corollary 3: The inequalities in (3) and (4) are exact opposites of each other, except for their qualification in the corollary text. It is not clear mathematically why these two inequalities are opposite.
5. Minimizing loss in (5): Algorithm 1 suggests that the action selection is not used during training time and only at inference time. Would not the policy benefit from doing BC training with this action selection method?
6. Weaker policy used for contrastive $\mathcal{L}_F$: using an older, weaker policy for a contrastive loss may lead to replacing some design decisions with equally harder decisions that must be made by the practitioner.
7. No error bars: Figure 6 display main results of the superiority of the authors' method but they do not include any statistical significance on the results they report. This is a major weakness in my opinion and indicates that the experimentation was not sufficient.

**Questions:**

In addition to addressing the weaknesses, I would like the authors to answer the following questions:
1. In equation (6), why use MSE style loss as opposed to a KL divergence on the action distributions? Is MSE more robust in this context?
2. Can the authors describe more technically why this methodology has a focus on human demonstrations? The argument of latent intentions is not clear to me.

---

> ### Author Response · Authors · 2024-11-22
>
> Thank you for your thoughtful feedback. Please find our response to your comments below.
>
> > Add error bars in Figure 6 for statistical significance.
> * We have re-run all methods on all seven simulation tasks across three seeds. The results with error bars are updated in Figure 6, Figure 7, and Tab 4. The new results confirm the statistical significance of the improvements provided by our method.
>
> > Difference between Corollary 2 and Corollary 3. It is not clear mathematically why these two inequalities are opposite.
> * The difference between the two corollaries arises from the environments considered.
>    * For corollary 2, the absence of noise implies $P_f \rightarrow 1$ in deterministic environments. By the definition of temporal dependency, $\alpha_b > 0$. Hence, the right-hand side of Eq (2) is strictly negative.
>     * For corollary 3, in the stochastic environment with high entropy, $P_b \rightarrow 0$. Under the condition that temporal dependency decreases over time, $\alpha_f > \epsilon_b$, the left-hand side of Eq (2) becomes strictly positive.
>
> > Ambiguous definitions in 3.2: the appendix provides better definitions of the $\alpha$ and $\epsilon$, but the definitions in the main paper are not sufficient.
> * In our updated manuscript, we have revised the definitions of these terms in Section 3.2 for greater clarity, keeping in line with the formal definition in Appendix D1.
>
> > In equation (6), why use MSE style loss as opposed to a KL divergence on the action distributions?
> * The action space is continuous in the context of robot manipulation, i.e. the model outputs continuous action vectors rather than probability distributions over discrete actions. MSE is a natural choice for measuring distances between continuous actions. We discuss alternative metrics in Appendix A6.
>
> > Lack of model-based/model-free discussion. In my understanding of the analysis, the authors expect the policies not to explicitly learn dynamics models.
> * Our work indeed focuses on the model-free approach, as this aligns with the state-of-the-art methods in robot learning from human demonstrations [1-4].
>
> > Weaker policy used for contrastive $\mathcal{L}_F$ may lead to hard decisions for practitioners.
> * While our method adds a layer of decision-making for practitioners, the added complexity is comparable to other decoding methods [5-7] and justified by the performance improvements demonstrated in our experiments (Fig 12 and Tab 4).
>
> > Can the authors describe why this methodology has a focus on human demonstrations? The argument of latent intentions is not clear to me.
> * Compared to scripted demonstrations, human demonstrations typically involve latent factors that (i) introduce dependencies over time (e.g., subgoals) and (i) vary across individuals and demonstrations (e.g., handedness). This informs the non-Markovian assumption in our analysis and motivates our design of BID to address the consistency-reactivity tradeoff.
>
> > Algorithm 1 suggests that the action selection is not used during training time and only at inference time. Would not the policy benefit from doing BC training with this method?
> * The focus of this work is on test-time decoding without modifying policy training. Nevertheless, extending our method to policy training can be an exciting avenue for future work, e.g. distilling the outcomes of bidirectional sample selection into a revision policy [8].
>
> [1] Diffusion Policy, 2023 \
> [2] Learning Fine-Grained Bimanual Manipulation with Low-Cost Hardware, 2023 \
> [3] VQ-BeT: Behavior Generation with Latent Actions, 2024 \
> [4] Pi0: A Vision-Language-Action Flow Model for General Robot Control, 2024 \
> [5] Contrastive Decoding: Open-ended Text Generation as Optimization, 2022 \
> [6] Fast Inference from Transformers via Speculative Decoding, 2023 \
> [7] Accelerating Large Language Model Decoding with Speculative Sampling, 2023 \
> [8] Recursive Introspection: Teaching Language Model Agents How to Self-Improve, 2024

---

> ### Author Response · Authors · 2024-11-26
> **Follow-Up**
>
> Dear Reviewer FR7d,
>
> Thank you again for your thoughtful review. As the first discussion phase nears its close, we wanted to check if you have any remaining questions or concerns.
>
> In our initial response, we assessed statistical significance (e.g., fig 6, fig 7, tab 4), clarified our analysis and methods (e.g., corollary, model-free, latent variables, distance metric), and revised the manuscript accordingly.
>
> We hope these updates have addressed your concerns. If there’s anything else we can further clarify, please let us know. Thanks!

---

> ### Comment · Reviewer_FR7d · 2024-11-27
> **Thank you for the clarifications.**
>
> Authors,
>
> Thank you for the clarifications. I believe they make the paper stronger and much clearer. I will maintain that the proofs corollaries 2 and 3 in the appendix are still wanting for more rigor; however, I do not believe they are critical to the main paper.

---

> ### Author Response · Authors · 2024-11-27
>
> Thank you for your encouraging feedback on our rebuttal! We’re glad to hear that our revisions have `made the paper stronger and much clearer`. We've further updated the manuscript with polished proofs in Appendix D5. Thank you again for your valuable suggestions!

---

### Official Review · Reviewer_PZsG · 2024-11-04

**Soundness:** 3
**Presentation:** 4
**Contribution:** 3
**Rating:** 6
**Confidence:** 4

**Summary:**

The paper introduces Bidirectional Decoding (BID), a test-time inference algorithm designed to improve action chunking in robotic policies by balancing long-term consistency with reactivity to unexpected changes in the environment. The authors address the tradeoff observed in action chunking, where increased chunk sizes help capture temporal dependencies but reduce the robot’s responsiveness to dynamic elements. BID operates by sampling multiple actions at each time step and selecting the optimal one based on two criteria: backward coherence (alignment with prior actions) and forward contrast (similarity to a stronger, more optimal policy and difference from a weaker one). The approach enables more flexible, closed-loop decision-making that adapts to real-time conditions. Experimental evaluations across multiple simulation benchmarks and real-world tasks show that BID consistently enhances the performance of robotic policies, demonstrating its effectiveness as a plug-and-play solution to improve the robustness and accuracy of action-chunked behavior cloning models in robotics.

**Strengths:**

1. The concept of "Action Chunking" is relatively new, and further research could reveal additional advantages of this technique by deepening our understanding of its potential benefits.

2. Experimental results demonstrate the promise of the proposed method, as it outperforms several state-of-the-art (SOTA) diffusion-policy-based behavior cloning techniques, suggesting it could be a viable approach to enhancing robotic policy learning.

**Weaknesses:**

1. From my understanding, Action Chunking primarily focuses on capturing implicit hierarchical relationships in action sequences without needing to acquire explicit representation of reusable skills or options (where "options" refers to temporally extended actions in hierarchical reinforcement or imitation learning). The field of option/skill discovery is active an active research direction, with an aim to explore methods to learn options/skills from either environmental interactions or recorded demonstrations.

2. Because Action Chunking requires pre-defined window sizes for past state observations and future actions, option/skill discovery methods (e.g. [1][2]) are potentially more adaptive and flexible than Action Chunking. It would be valuable for the paper to discuss and compare Action Chunking theoretically and experimentally with option/skill discovery approaches. Notably, in hierarchical RL, a common approach involves an initial option discovery phase, followed by a reinforcement learning phase utilizing these options. In this work, imitation learning could replace the second phase. Two recent, notable works on option/skill discovery [1][2] could serve as relevant comparisons.

3. Another limitation is that the reported performance improvements (e.g., Table 1) are modest. For example, the difference between the results of EMA (Zhao et al., 2023) in the 3rd row and the proposed method in the 4th row is relatively small. This pattern holds in Tables 2 and 3 as well: when compared only against the weakest baseline (Vanilla), the performance gains remain minor.

4. Furthermore, Tables 2 and 3 omit results for other leading baselines, such as Warmstart (Janner et al., 2022) and EMA (Zhao et al., 2023), which would strengthen the evaluation of the proposed method.

References:

[1] Bagaria, A., & Konidaris, G. (2019, September). Option discovery using deep skill chaining. In International Conference on Learning Representations.

[2] Zhao, Tianxiang, et al. "Skill disentanglement for imitation learning from suboptimal demonstrations." Proceedings of the 29th ACM SIGKDD Conference on Knowledge Discovery and Data Mining. 2023.

**Questions:**

Why is it necessary to set the 𝑐 parameter, which represents the window size of incorporated historical states? When employing an LSTM-based policy, the incorporation of historical states becomes more flexible. This flexibility is akin to the context window size used in word2vec for planning. The advantages of using LSTMs over a fixed context window 𝑐 have been discussed in the following paper: [1].

Reference:

[1] Zhuo, Hankz Hankui, et al. "Discovering underlying plans based on shallow models." ACM Transactions on Intelligent Systems and Technology (TIST) 11.2 (2020): 1-30.

---

> ### Author Response · Authors · 2024-11-22
>
> Thank you for your thoughtful feedback. Please find our response to your comments below.
>
> > Performance improvements are modest. For example, the difference between the results of EMA and the proposed method is relatively small.
> * As summarized in Fig 7, our method yields a 32% relative performance gain, which is more than double the 15% improvement obtained with EMA.
> * Additionally, in scenarios where sequential chunks diverge into different strategies or the prior chunk is suboptimal in stochastic environments, EMA may lead to detrimental effects, as discussed in Appendix A4 & A5. In comparison, our proposed BID is robust across different tasks and conditions, as shown in Fig 6 and Tab 4.
> * Moreover, EMA and our method are complementary rather than competing approaches. When combined, they result in even stronger performance, yielding a 46% relative improvement averaged across seven simulation tasks.
>
> > Tables 2 and 3 omit results for other baselines, such as Warmstart and EMA, which would strengthen the evaluation of the proposed method.
> * We have extended our experiments to EMA on VQ-BeT. The result, summarized in the table below, confirms the advantage of BID over EMA, particularly under stochastic conditions.
> * Regarding the Warmstart baseline, to our knowledge, it is a sample initialization technique tailored to diffusion-based policies. If there exists an analogous formulation for VQ-BeT, we would greatly appreciate a point of reference, and we would be happy to further extend our evaluations.
> * We have included the new experimental results in Appendix A5 of the updated manuscript.
>
>   | Stochastic Noise | 0.0  | 1.0 | 1.5  | Average |
>   |------------------|--------------|--------------|--------------|-----------------|
>   | Open-Loop Vanilla | 64.0 ± 4.2   | 26.9 ± 2.8   | 13.0 ± 0.4   | 34.6 ± 1.7      |
>   | Open-Loop EMA | 64.1 ± 1.7   | 27.6 ± 3.3   | 12.9 ± 1.1   | 34.9 ± 1.3      |
>   | Open-Loop BID (ours) | **66.1 ± 3.5**  | 31.4 ± 3.0   | 16.0 ± 1.2   | 37.8 ± 1.6      |
>   | Closed-Loop Vanilla | 48.9 ± 2.7   | 38.3 ± 3.4   | 29.5 ± 0.9   | 38.9 ± 1.5      |
>   | Closed-Loop EMA  | 52.6 ± 2.9   | 35.7 ± 2.2   | 18.4 ± 2.3   | 35.6 ± 1.4      |
>   | Closed-Loop BID (ours) | 54.4 ± 1.8   | **45.3 ± 3.8**  | **31.7 ± 0.3** | **43.8 ± 1.4** |
>
>
> > From my understanding, action chunking primarily focuses on capturing implicit hierarchical relationships in action sequences. It would be valuable to discuss and compare action chunking with option/skill discovery approaches.
> * Action chunking and option discovery indeed share similarities in modeling temporally extended actions. However, their motivations, designs and practical outcomes are often different:
>   * Option discovery typically aims to learn hierarchical policies, explicitly discovering high-level skills from low-level action sequences. This approach is conceptually compelling, but still faces challenges in scaling to large robotic datasets.
>   * In contrast, action chunking is designed to group multiple low-level actions into a chunk decision. Each chunk does not necessarily correspond to a high-level skill, but serves as a practical means to capture temporal consistency, as analyzed in Sec 3.2.
> * In the updated manuscript, we have added a detailed discussion in Appendix C, elaborating on the connections and distinctions between action chunking and option discovery [1-4]. We would be happy to provide further clarifications or comparisons if needed.
>
> > Why is it necessary to set the 𝑐 parameter? LSTM-based policy is more flexible.
> * LSTMs are indeed more flexible in modeling long-term dependencies. However, in the context of imitation learning, extending the context length may lead to decreased performance due to spurious correlations, as ablated in Appendix A2 and studied in prior work [5,6].
> * To mitigate the robustness issues, recent methods in imitation learning from human demonstrations often use a small, fixed context length (e.g., 1 [7]). We follow this convention and set the $c$ parameter accordingly.
> * To clarify this, we have added discussions on the LSTM-based policy in Appendix C.
>
> [1] Option discovery using deep skill chaining. 2019 \
> [2] Learning Options via Compression, 2022 \
> [3] Skill disentanglement for imitation learning from suboptimal demonstrations. 2023 \
> [4] FedSkill: Privacy Preserved Interpretable Skill Learning via Imitation, 2023 \
> [5] Causal Confusion in Imitation Learning, 2019 \
> [6] Fighting CopyCat Agents Behavioral Cloning from Observation Histories, 2020 \
> [7] OpenVLA:An Open-Source Vision-Language-Action Model, 2024

---

> ### Comment · Reviewer_PZsG · 2024-11-25
> **Replying to the Authors**
>
> Thank you for your detailed response and the additional experiments. I appreciate the clarifications and the updates addressing my earlier concerns. While I find the results regarding performance improvements (question 1) satisfactory, I remain unconvinced by some of the responses to other points, which I elaborate on below.
>
> **Option Discovery vs. Action Chunking:**
>
> The claim that option discovery faces scalability issues with large robotic datasets, as stated in *“Option discovery typically aims to learn hierarchical policies … faces challenges in scaling to large robotic datasets”*, does not appear well-supported. Both approaches can leverage the same datasets, and there does not seem to be an inherent limitation in applying option discovery frameworks to large-scale datasets with robot trajectories. Since your robotic datasets already include information about trajectory success or failure, this data could be directly used to label trajectories for option discovery.
>
> In responding to *“In contrast, action chunking is designed to couple multiple low-level actions into a coherent chunk. Each chunk does not necessarily correspond to a high-level skill, but serves as a practical means to capture temporal consistency”*:
>
> Option discovery is explicitly designed to extract high-level abstractions to a formal format, which might offer even better modeling of temporal consistency than action chunking. While you argue that action chunking provides practical advantages, both methods fundamentally address temporal dependencies. Option discovery, however, has the added benefit of theoretical guarantees rooted in Semi-MDPs and planning, making it a potentially more robust and flexible approach.
>
> Overall, experimental comparisons between option discovery and action chunking would be crucial to demonstrate the practical advantages of your approach and validate its novelty.
>
> **Responding to “*LSTMs are indeed more flexible in modeling long-term dependencies. However, in the context of imitation learning, extending the context length can lead to decreased performance due to spurious correlations*”:**
>
> LSTMs, or Long Short-Term Memory networks, are explicitly designed to learn what information to retain and what to forget at each step, enabling them to effectively handle both long-term and short-term dependencies. While you are correct that LSTMs are good at capturing long-term dependencies, their architecture inherently allows them to model short-term dependencies as well, enabling them to flexibly adapt to various temporal patterns. This flexibility means that LSTMs can dynamically adjust between modeling long- and short-term relationships based on the dataset and the training signal, optimizing their strategy for the specific context. To strengthen your claims, I believe experiments incorporating LSTM-based policies with flexible horizons would be valuable in demonstrating whether fixed context lengths truly provide a performance advantage.

---

> ### Author Response · Authors · 2024-11-26
>
> Thank you for your detailed feedback on our rebuttal!
>
> We appreciate your encouraging comment on **`performance improvements satisfactory`**, and we agree on the advantages of `option discovery` and `lstm`, e.g., `flexible` acknowledged in our initial response and `theoretical roots` noted in your comments.
>
> That said, we believe the *additional experiments requested on these topics are beyond the scope of this work*, as we clarify below.
>
> - **Core contributions**: our paper focuses specifically on **understanding and improving action chunking, without modifying training methods or model architectures** (e.g., fixed chunk size, short context length, feedforward encoder) used in state-of-the-art robot policies learned from human demonstrations [1-5]. Our key contributions are:
>     - **first action chunking analysis**: reveal the reason behind the conflicting observations in recent literature
>     - **first action decoding method**: address the limitation of action chunking with additional test-time compute
>
> - **Experiment designs**: to validate these contributions, we have conducted a rich set of experiments across *widely used action chunking setups*:
>     - **2 policy classes**: diffusion policy and vq-bet
>     - **3 inference baselines**: vanilla, warmstart, ema
>     - **10 experimental settings**: 1 synthetic, 7 simulated, 2 real-world
>     - **extensive ablation studies**: Appendix A2, A3, A4, A5, A6
>
> - **Response clarifications**: our previous response was not meant to understate the potential of `option discovery` and `lstm`, but rather to explain our focus
>     - action chunking has been scaled to robotic foundation models but still lacks a thorough understanding
>     - we acknowledge that `option discovery` and `lstm` hold promise to supersede action chunking in the future, and have updated Appendix C to reflect this discussion
>
> - **Factual corrections**: finally, we would like to address two *potential misunderstandings*:
>     - `your robotic datasets already include information about trajectory success or failure`: this is not true, e.g., all trajectories in BC simulation benchmarks (Sec 5.2) are successful demonstrations
>     - `LSTMs handle both long-term and short-term dependencies`: our previous response did not dispute the modeling strengths of LSTM. It aimed to answer your question on the context length `𝑐 parameter`, which is often kept small in BC (e.g., 1 in OpenVLA, 2 or 3 in $\pi_0$) to prevent exploiting spurious correlations between previous and future actions. This robustness challenge arises from training data and learning algorithms, not architectures (please refer to `causal fusion` and `copycat agent` in our previous response).
>
> We hope these help address your concerns about experimental comparisons. Thank you again for your valuable feedback and your engagement with our submission.
>
> [1] Diffusion Policy, 2023 \
> [2] Learning Fine-Grained Bimanual Manipulation with Low-Cost Hardware, 2023 \
> [3] VQ-BeT: Behavior Generation with Latent Actions, 2024 \
> [4] Octo: An Open-Source Generalist Robot Policy, 2024 \
> [5] π0: A Vision-Language-Action Flow Model for General Robot Control, 2024

---

> ### Author Response · Authors · 2024-11-28
> **Follow-Up**
>
> Dear Reviewer PZsG,
>
> We would like to follow up on our discussions and check if our previous response has addressed your concerns.
>
> To briefly recap:
> - technical contributions: (i) understand and (ii) improve action chunking, *without model modifications*
> - empirical evaluations: 2 sota policies, 3 recent baselines, 10 different settings, 5 appendix ablations
> - average outcomes: +32% independently, +46% when combined with ema, `improvements satisfactory`
>
> Due to time constraints during the first discussion phase, we could not add implementation on new codebases, but we have added discussions on the potential advantages of 'option discovery' and 'lstm' in Appendix D.
>
> Could you please let us know if these updates sufficiently address your concerns about experiments, with respect to our claimed technical contributions?
>
> We are willing to address any further concerns during the second discussion phase. Thanks again for your time and valuable feedback!

---

> ### Author Response · Authors · 2024-12-01
> **Second Follow-Up**
>
> Dear Reviewer PZsG,
>
> We wanted to kindly follow up again on our previous discussions.
>
> Could you please let us know if you have any remaining critical concerns regarding the experiments, with respect to our *two claimed technical contributions* (achieved without model modifications)?
>
> Thank you again for your time and valuable feedback.

---

> > ### Comment · Reviewer_PZsG · 2024-12-03
> >
> > Thank you for your response! I’m happy to increase my score.

---

### Official Review · Reviewer_ayXv · 2024-11-11

**Soundness:** 3
**Presentation:** 3
**Contribution:** 2
**Rating:** 6
**Confidence:** 4

**Summary:**

This paper examines the impact of action chunking in robot learning from demonstrations, where actions are executed in sequences without intermediate replanning. While action chunking helps robots learn temporal dependencies, it can hinder adaptability in dynamic environments. To mitigate this tradeoff, the authors propose Bidirectional Decoding (BID), a test-time inference method that enhances reactivity while preserving the benefits of chunking. BID samples multiple actions per step, selecting the best based on backward coherence (consistency with prior actions) and forward contrast (alignment with stronger policy predictions and deviation from weaker ones). This approach improves temporal consistency and adaptability, yielding better results across multiple simulation benchmarks and real-world tasks. On the down-side, the approach requires significantly more compute as multiple model outputs are sampled. Authors deal with that by parallelizing the computation.

**Strengths:**

- Proposed method seems reasonable and well-thought of, with clear description and theoretical backing.
- Paper is well written, clear and well-structured. The motivation and the idea of the paper is clear. The approach is clearly stated and all relevant background is introduced well. The method explanation is accompanied with appropriate visualisation figures that help understanding.
- The proposed method achieves good performance on target experiments.
- The approach is tested in several simulation and real experiments.

**Weaknesses:**

- The method requires significantly more compute than the base methods.
- The definition of a horizon is sometimes unclear. In Moving Horizon fields (e.g. Model Predictive Control - MPC) horizon is what is presented in the paper as $l$. Perhaps this should be also labeled on the Figure 3. The horizon $h$ in this paper is used to describe what is replanning time in MPC.
- Generally, it would be useful to back this more to the MPC field. As focus of the paper is not on the inference itself but more on using any planning/prediction model in the closed-loop. Borelli, Bemporad, Morari, Predictive control book.
- The analysis on the replanning times and prediction horizon seems to be a bit limited and not fully consistent among different ablation studies.
- A single experimental problem is used for analysing the main point of the paper (demonstration with iddle action).  Figure 5 is not clear, why the results for 3 are better than for 5.
- Backward coherence as introduced here is a standard for improving temporal consistency when replanning in many MPC approaches.
- Description of policies in 3.2. is a bit confusing. It is not exactly clear are they aligned in time and why the importance is not same for the context for the first frame before the predicitons.
- In line 316 and 318, it is stated “equally optimal” and “the most optimal” that is in contradiction with the definition of the optimality.
- Using negative sets is not motivated well. This could be also investigated with ablation studies.

**Questions:**

- Can you clarify the definition of horizon, and relationship to MPC.
- Can you clarify experiments  in section 3.2.
- Can you explain in more detail dirrente policies for analysis in 3.2. E.g. can you show in Figure 3 with aligned current time. For time $t_k$ you predict the sequence with some length and take context with some length.
- In Corollary 2 you claim that $\pi_{h+d}$ is more disadvantageous but you show in eq (2) that the expected loss is smaller. This seems to be contradictory.
- Why is using negative set important?
- I would suggest using bar plot for fig 6 right.

---

> ### Author Response · Authors · 2024-11-22
>
> Thank you for your thoughtful feedback. Please find our response to your comments below.
>
> > Can you clarify the definition of horizon, and relationship to MPC?
>
> * Prediction horizon $l$ refers to the number of future actions predicted as a single chunk; action horizon $h \leq l$ refers to the number of actions executed from the chunk before replanning. These definitions align with recent works in BC from human demonstrations [1] and are analogous to the prediction horizon and control horizon in MPC.
> * A crucial distinction between traditional MPC and recent BC lies in the choice of action/control horizon. MPC typically uses a short control horizon (e.g., h=1), whereas recent BC policies use longer action horizons (e.g., h=8 [1] or h=50 [2]).
>
> > A single experimental problem is used for analysis (demonstration with idle action).
> * We conducted *two experiments* to validate our analysis: synthetic 1d problems (idle actions, Fig 5) and simulated robotic tasks (action noise, Tab 1). To improve clarity, we have revised the Sec 5.2.1 to explicitly reference both experiments and their roles in supporting our analysis.
>
> > The analysis on the replanning times and prediction horizon seems to be not fully consistent among different ablation studies.
> * Throughout our analysis and ablation studies, we focus on the action horizon, as it is the key parameter influencing the consistency-reactivity tradeoff. To enhance clarity, we have revised A2 in our manuscript.
>
> > Figure 5 is not clear, why the results for 3 are better than for 5.
> * Figure 5 is designed to validate our theoretical findings by comparing policies across different action horizons. Our results demonstrate that long-horizon learners align more closely with the long-idling expert in deterministic environments, while short-horizon learners align better with the short-idling expert in stochastic settings. Intermediate horizons (e.g., ah=5) exhibit a trade-off between these two extremes.
> * To improve visual clarity, we have revised Fig 5 in our updated manuscript, including (i) replacing the KDE plot with separate histograms to facilitate direct comparison, (ii) visualizing only the results for horizons 1, 5, and 10 to enhance readability. Detailed results for all horizons are provided in A1.
>
> > Can you explain in more detail different policies for analysis in 3.2. E.g. can you show in Figure 3 with aligned current time?
> * In Sec 3.2, we consider three types of policies: a long-context expert $\pi_{(k,1)}$, a short-horizon learner $\pi_{(c,h)}$ and a long-horizon learner $\pi_{(c,h+d)}$. These policies differ in the information they directly observe from the context and the information they can infer from the predicted action chunk.
> * We have revised the Figure 3 in our update manuscript, annotating the current time step and visualizing the discarded predictions resulting from replanning.
>
> > In Corollary 2 you claim that $\pi_{h+d}$ is more disadvantageous but you show in eq (2) that the expected loss is smaller. This seems to be contradictory.
> * Corollary 2 does not claim $\pi_{h+d}$ is more disadvantageous. Instead, we show that it is more advantageous in deterministic environments, as the expected loss is lower in Eq (3).
> * This can be seen from the right-hand side of Eq (2). In deterministic settings, where $P_f \approx 1$ (absence of noise) and $\alpha_b > 0$ (by definition), the RHS is strictly negative, leading to a reduction in expected loss.
>
> > Using negative sets is not motivated well. This could be also investigated with ablation studies.
> * The motivation is to mitigate the discrepancy between the learned policy and the expert policy. As illustrated in Fig 10, by contrasting against negative samples drawn from a weaker policy, our method encourages the selected sample to be closer to high-likelihood expert behaviors.
> * In our initial manuscript, we discussed this motivation in Appendix B and provided an ablation study on Diffusion Policy in Appendix A3. To further clarify this motivation, we have expanded the discussion in the main draft (Sec 4.3) and expanded the ablation study to VQ-BET (A5).
>
> > The method requires significantly more computation than the base methods.
> * Our method indeed requires more computation. However, the computation is fully parallelizable. As a result, the runtime overhead is modest, ~2x compared to other closed-loop baselines, as shown in Tab 2.
> * Moreover, we view this limitation as an opportunity for future research. As discussed in Sec 6, potential avenues for addressing this include guided sampling [3], revision distillation [4], alongside hardware accelerations.
>
> > I would suggest using bar plot for Fig 6 right.
> * Thank you for the suggestion! We have updated Fig 6 & 7 accordingly.
>
> [1] Diffusion Policy, 2023 \
> [2] Pi0: A Vision-Language-Action Flow Model for General Robot Control, 2024 \
> [3] Classifier-free diffusion guidance, 2022 \
> [4] Recursive Introspection: Teaching Language Model Agents How to Self-Improve, 2024

---

> ### Author Response · Authors · 2024-11-26
> **Follow-Up**
>
> Dear Reviewer ayXv,
>
> Thank you again for your thoughtful review. As the first discussion phase nears its close, we wanted to check if you have any remaining questions or concerns.
>
> In our initial response, we
> - provided clarifications on *key comments* (e.g., relation to MPC, horizon analysis, negative set, corollary insights, computation cost)
> - revised our manuscript to address *potential misunderstandings* (e.g., illustrations, diagnostic experiments, bar plot, ablation studies)
>
> We believe our manuscript has been considerably improved by incorporating your feedback. We look forward to your reassessment and are willing to address any other concerns. Thanks!

---

> > ### Comment · Reviewer_ayXv · 2024-11-28
> > **Answer to authors**
> >
> > I would like to thank the authors for their elaborate response and for addressing my concerns. Some of my concerns are addressed, except the first concern regarding the relationship with MPC (Moving Horizon Control). I believe this connection should be significantly more elaborated as the focus of the paper is exactly on this, the prediction horizon length, ‘frequency’, and the context length. I believe that is an important aspect of the readability of the work and setting up the baseline for the follow-up research in this direction. Relevant literature in the MPC should be addressed. And notation used in this paper should be in line with the other literature. In  [2] as well as in [3], there is a single horizon for action chunks.  In [1], authors explicitly states that they employ receding horizon control with clear distinction of observation, prediction and action horizons.  Here the “horizon” is used interchangeably with “action horizon” from [1]. This is not appropriate.
> >
> > If I am not mistaken, Figure 3 presents the worst-case scenario for a given (c, h) policy.  It is still not clear, and the prediction horizon should also be presented there.
> >
> > The results from figure 5 seem to be affected by the prediction horizon length. The context $c$ and action horizon $h$ are very important, but also prediction horizon $l$ should be analyzed. There are a lot of issues if $h$ is close to the $l$. Of course, it is not possible to directly compare this approach to the receding horizon control approaches and they are mainly model-based [4].
> >
> > However, I believe this is an interesting read, and these issues can be resolved for the final version, so I can increase my rating.
> >
> > [1] Diffusion Policy, 2023
> >
> > [2] Pi0: A Vision-Language-Action Flow Model for General Robot Control, 2024
> >
> > [3] Zhao, Tony Z., Vikash Kumar, Sergey Levine, and Chelsea Finn. "Learning Fine-Grained Bimanual Manipulation with Low-Cost Hardware.”
> >
> > [4]Oops! I cannot do it again: Testing for recursive feasibility in MPC

---

> ### Author Response · Authors · 2024-11-29
>
> Dear Reviewer ayXv,
>
> Thank you for your thoughtful response to our rebuttal! Below are our brief responses to your key comments:
> - *Re*: `horizon is used interchangeably with action horizon, this is not appropriate`:\
>   We use "horizon" as shorthand for the action horizon $h$, as our analysis explicitly focuses on $h$, assuming a sufficiently long prediction length $l$.
> - *Re*: `notation used in this paper should be in line with the other literature`:\
>   Our notations are aligned with that in DP and VQ-BeT. We differ from $\pi_0$ and ACT because, as you noted, those works do not distinguish $h$ and $l$.
> - *Re*: `the results from figure 5 seem to be affected by the prediction length`:\
>   This is not the case. Our synthetic experiment specifically validates the influence of action horizon $h$, with all policies using the same prediction length $l$.
>
>
> We will make sure to further clarify our notations, elaborate on connections to MPC, and incorporate the relevant literature you suggested in the final manuscript.
>
> Thank you again for your valuable feedback, which has been instrumental in improving the clarity of our paper.

---

### Author Response · Authors · 2024-11-22
**General Response and Manuscript Updates**

We thank all the reviewers for the thoughtful feedback, which have helped to improve our paper.

Below, we outline the key updates to our manuscript, which address the following:
* **Expanded simulation experiments**: repeated all simulations with three random seeds, updated relevant plots and tables in Sec 5.2, verified the statistical significance of our findings.
* **Ablation studies on VQ-BET**: validated the importance of each proposed component, confirmed the advantage of BID over the other competing baselines and variants.
* **Alternative distance metrics**: examined the sensitivity of BID to the choice of distance metric, demonstrated its effectiveness with both L2 and L1 distances.
* **Manuscript clarifications**: revised visual presentations for Figure 3 & 5, clarified setup and definitions in Sec 3.2, expanded the discussion of our method in Sec 4.3, and incorporated additional related work in Appendix C.

---

### Meta-Review · Area_Chair_jf3D · 2024-12-19

**Metareview:**

The paper studies the effect of action chunking on robot learning from demonstrations. It proposed BID (Bidirectional Decoding), a test-time sampling procedure that aims to make standard action chunking more reactive. BID samples multiple actions at each time step and selects the best based on `backward coherence` and `forward contrast.` Experiments showing the usefulness of such an approach are conducted in sim and real world.

Strengths:
+ Paper is well-written
+ Method is well thought of
+ Experiments demonstrate promising results

Weakness:
+ More compute than standard action chunking

All reviewers recommend acceptance and the AC agrees. The method is well thought out, and the results are promising.

**Additional Comments On Reviewer Discussion:**

All reviewers asked clarificatory questions that were well addressed by the authors. Besides the clarifications, Reviewer ayXv felt the connection to MPC could be better elaborated, which the authors agreed to do in the final manuscript. Reviewer FR7d  maintained that the proofs corollaries 2 and 3 in the appendix required more rigor but believed they were not critical for the paper.

---

### Decision · Program_Chairs · 2025-01-22

Accept (Poster)